



# Organic and inorganic bromine measurements around the extratropical tropopause and lowermost stratosphere: Insights into the transport pathways and total bromine

Meike K. Rotermund[1], Vera Bense[2], Martyn P. Chipperfield[3,4], Andreas Engel[5], Jens-Uwe Grooß[6], Peter Hoor[2], Tilman Hüneke[1,a], Timo Keber[5], Flora Kluge[1], Benjamin Schreiner[1], Tanja Schuck[5], Bärbel Vogel[6], Andreas Zahn[7], and Klaus Pfeilsticker[1]

[1]Institute of Environmental Physics, University of Heidelberg, Heidelberg, Germany
[2]Institute for Atmospheric Physics, Johannes Gutenberg University Mainz, Mainz, Germany
[3]School of Earth and Environment, University of Leeds, Leeds, UK
[4]National Centre for Earth Observation, University of Leeds, Leeds, UK
[5]Institute for Atmospheric and Environmental Science, Goethe University Frankfurt, Frankfurt, Germany
[6]Institute of Energy and Climate Research - Stratosphere (IEK-7), Forschungszentrum Jülich, Jülich, Germany
[7]Institute for Meteorology and Climate Research, Karlsruhe Institute of Technology (KIT), Karlsruhe, Germany
[a]now with Encavis AG, Hamburg, Germany

**Correspondence:** Meike Rotermund (Meike.Rotermund@iup.uni-heidelberg.de)

**Abstract.** We report on measurements of total bromine ($Br^{tot}$) in the upper troposphere and lower stratosphere taken during 15 flights with the German High Altitude and LOng range research aircraft (HALO). The research campaign WISE (Wave-driven ISentropic Exchange) included regions over the North Atlantic, Norwegian Sea and north-western Europe in fall 2017. $Br^{tot}$ is calculated from measured total organic bromine ($Br^{org}$) added to inorganic bromine ($Br_y^{inorg}$), evaluated from measured $BrO$

and photochemical modelling. Combining these data, the weighted-mean $[Br^{tot}]$ is $19.2 \pm 1.2$ ppt in the northern hemispheric lower stratosphere (LS) in agreement with expectations for $Br^{tot}$ in the middle stratosphere (Engel and Rigby et al. (2018)). The data reflects the expected variability in $Br^{tot}$ in the LS due to variable influx of shorter-lived brominated source and product gases from different regions of entry. A closer look into $Br^{org}$ and $Br_y^{inorg}$, as well as simultaneously measured transport tracers ($CO$ and $N_2O$) and an air mass lag-time tracer ($SF_6$), suggests that bromine-rich air masses persistently protruded into

the lowermost stratosphere (LMS) in boreal summer, creating a high bromine region (HBrR). A subsection, HBrR*, has a weighted average of $[Br^{tot}]=20.9 \pm 0.8$ ppt. The most probable source region is former air from the tropical upper troposphere and tropopause layer (UT/TTL) with a weighted mean $[Br^{tot}]=21.6 \pm 0.7$ ppt. CLaMS Lagrangian transport modelling shows that the HBrR air mass consists of $51.2\%$ from the tropical troposphere, $27.1\%$ from the stratospheric background, and $6.4\%$ from the mid-latitude troposphere (as well as contributions from other domains). The majority of the surface air reaching the

HBrR is from the Asian monsoon and its adjacent tropical regions, which greatly influences trace gas transport into the LMS in boreal summer and fall. Tropical cyclones from Central America in addition to air associated with the Asian monsoon region contribute to the elevated $Br^{tot}$ observed in the UT/TTL. TOMCAT global 3–D model simulations of a concurrent increase of $Br^{tot}$ show an associated $O_3$ change of $-2.6 \pm 0.7\%$ in the LS and $-3.1 \pm 0.7\%$ near the tropopause. Our study of varying $Br^{tot}$ in the LS also emphasizes the need for more extensive monitoring of stratospheric $Br^{tot}$ globally and seasonally to fully





understand its impact on LMS $O_3$ and its radiative forcing of climate, as well as in aged air in the middle stratosphere to elucidate the stratospheric trend in bromine.

# 1 Introduction

Bromine is thought to account for approximately one third of the global stratospheric $O_3$ depletion, and it is particularly effective at destroying $O_3$ in the lowermost stratosphere (LMS) (e.g., Salawitch et al. (2005); Hossaini et al. (2015); WMO
(2018); Chipperfield et al. (2020)). The LMS is defined as the atmospheric layer extending from the local tropopause (TP) to the potential temperature $\Theta = 380$ K isentrope thereby excluding the tropics (Holton et al., 1995), and we define the lower stratosphere (LS) from the local TP up to the highest measurements of this study at $\Theta \sim 410$ K. In the definition of the tropical tropopause layer (TTL), we follow Fueglistaler et al. (2009), i.e., for latitudes $< \sim 27°$ N and S and potential temperatures $\Theta = 355 - 425$ K.

There are four major contributions to total bromine (hereafter called $Br^{tot}$) in the troposphere as suggested by previous research (e.g., Engel and Rigby et al. (2018)): (1) $CH_3Br$ that has both natural and anthropogenic sources with a contribution of 6.8 ppt to $Br^{tot}$ in 2016, (2) the four major halons consisting of H–1211 ($CClBrF_2$), H–1301 ($CBrF_3$), H–1202 ($CBr_2F_2$), and H–2402 ($CBrF_2CBrF_2$) all from anthropogenic activities and contributing 7.8 ppt to $Br^{tot}$ in 2016, (3) bromine compounds inferred from naturally emitted so-called very short-lived substances ($Br^{VSLS}$), and (4) inorganic bromine previously released
from brominated VSLS or sea salt of which a fraction of both is eventually also transported into the stratosphere. Total organic bromine ($Br^{org}$) is the sum of contributions (1)–(3), while inorganic bromine defined here as $Br_y^{inorg}$ (often referred to as $Br_y$) consists of contribution (4).

The monitoring of $CH_3Br$ (e.g., Montzka et al. (2003)) and the longer lived halons (e.g., Vollmer et al. (2016)) is regularly performed by different observational networks, notably at NOAA (National Oceanic and Atmospheric Administration), UEA
(University of East Anglia), and AGAGE (Advanced Global Atmospheric Gases Experiment; Prinn et al. (2018)), amongst others. Numerous observational and theoretical studies have addressed the variable contributions (3)–(4), which together are assessed to contribute $5 \pm 2$ ppt to stratospheric bromine (e.g., Engel and Rigby et al. (2018) and references therein). These include studies performed from the ground, ships, manned and unmanned aircrafts, and high-flying balloons at different locations/seasons and globally from satellites. In order to answer the question 'how much bromine is in the stratosphere?', detailed
studies performed at the stratospheric entry in different regions and seasons are most suitable. However, additional studies addressing $Br^{tot}$ measurements in aged air of the middle and upper stratosphere at a given location and season are also crucial.

Many studies have occurred primarily in the tropics and the Asian monsoon during the convective season where most of the stratospheric air is believed to enter (e.g., Holton et al. (1995); Fueglistaler et al. (2004) and Ploeger et al. (2015)). Observational and theoretical studies addressing bromine at the stratospheric entry level in the tropics started well over two
decades ago. Numerous dedicated research activities have taken place to investigate the fate of atmospheric bromine (e.g., Jensen et al. (2013); Harris et al. (2017) and Pan et al. (2017)).



It is well known that contribution (3) to $\mathrm{Br^{tot}}$, the brominated VSLS summed-up according to their Br atomicity ($\mathrm{Br^{VSLS}}$), is predominantly from marine emissions of $CH_2Br_2$ and $CHBr_3$ with small contributions from other brominated hydrochloro-bromocarbons ($CH_2ClBr$, $CHCl_2Br$, $CHClBr_2$) (Engel and Rigby et al. (2018)). Early measurements of $CH_2Br_2$ and $CHBr_3$

at the tropical TP by Schauffler et al. (1993, 1998, 1999) indicated that both species may contribute substantially to $\mathrm{Br^{tot}}$ (e.g., a few ppts). More recent studies reported varying contributions of brominated VSLS in the tropical upper troposphere (UT) and TTL. The $\mathrm{Br^{VSLS}}$ observations range from $\sim 1-4$ ppt in the tropical UT/TTL (e.g., Laube et al. (2008); Brinckmann et al. (2012); Sala et al. (2014); Wisher et al. (2014); Navarro et al. (2015); Wales et al. (2018); Engel and Rigby et al. (2018) and Adcock et al. (2021)). A recent study by Keber et al. (2020), reported on aircraft measurements at the TP and in the LMS at

mid and high northern latitudes during different seasons. They observed $[\mathrm{Br^{VSLS}}] = 3.99 \pm 1.15$ ppt in the 10 K below the TP in boreal late summer and fall 2012/2017 and $[\mathrm{Br^{VSLS}}] = 5.20 \pm 1.25$ ppt in Dec. 2015–Mar. 2016. Keber et al. (2020) ascribed the varying mixing ratios found for the major brominated VSLS at the local TP to the location and seasonality of their source strengths, transport pathways and atmospheric lifetimes.

Some observations could also separately establish the magnitude of $\mathrm{Br_y^{inorg}}$ injection (contribution (4) to $\mathrm{Br^{tot}}$) into the

stratosphere. Namely Dorf et al. (2008) inferred $[\mathrm{Br_y^{inorg}}] = 2.1 \pm 2.6$ ppt (over Kiruna in 1999) and $4.0 \pm 2.5$ ppt (over north-eastern Brazil in June 2005) from simultaneous stratospheric balloon soundings of $\mathrm{Br^{org}}$ and $\mathrm{Br_y^{inorg}}$. Werner et al. (2017) found $[\mathrm{Br_y^{inorg}}] = 2.63 \pm 1.04$ ppt in the TTL over the tropical eastern Pacific during the convective season in 2013. Koenig et al. (2017) inferred $[\mathrm{Br_y^{inorg}}] = 2.3-3.1$ ppt in aged air masses in the TTL, and Wales et al. (2018) determined contribution (4) entering the stratosphere to be $2.1 \pm 2.1$ ppt for the western Pacific during the convective season in 2014.

Probing the middle and upper stratosphere for $\mathrm{Br_y^{inorg}}$ relies on measurements of the major inorganic bromine species BrO and/or $BrONO_2$ and suitable photochemical corrections for their ratio to $\mathrm{Br_y^{inorg}}$. These measurements can occur from the ground (e.g., Hendrick et al. (2007, 2008), and others), from balloons (e.g., Harder et al. (1998, 2000); Pfeilsticker et al. (2000); Dorf et al. (2006b, 2008); Höpfner et al. (2009); Stachnik et al. (2013); Kreycy et al. (2013) and Wetzel et al. (2017)), and/or satellite platforms (e.g., Sinnhuber et al. (2005); Sioris et al. (2006); Hendrick et al. (2009); Rozanov et al. (2011) and

Parrella et al. (2013)). All these studies, though their accuracy varies, point to the following: the contribution from (3) and (4) in the upper stratosphere ranges from $3-7$ ppt. Further, a decreasing trend of $\mathrm{Br^{tot}}$, reconcilable with decreasing contributions from (1) and (2), is discernable in aged stratospheric air until 2013 when the last balloon-borne $\mathrm{Br^{tot}}$ measurements in the middle stratosphere were reported (Fig. 1-16 of Engel and Rigby et al. (2018)).

$\mathrm{Br^{tot}}$ and reactive bromines are even less explored in the LMS in mid-latitudes for all seasons and the high latitude LS in

boreal summer. This region is particularly interesting since it comprises about $40\%$ of the stratospheric background air, though dynamically somewhat different from the tropical LS and 'overworld' circulation of the global middle stratosphere (e.g., Holton et al. (1995); Bönisch et al. (2009); Birner and Bönisch (2011); Gettelman et al. (2011); Ploeger et al. (2015) and Konopka et al. (2015)). It has recently attracted renewed interest since an apparent ongoing $O_3$ decrease is observed there, particularly in the northern hemisphere (NH) (e.g., Ball et al. (2018, 2019, 2020); Chipperfield et al. (2018, 2020)). The continuing negative

LS $O_3$ trends may partly be related to increases of anthropogenic and natural VSLS and their inorganic products (e.g., Ball et al. (2018); Barrera et al. (2020)). Simulations by Barrera et al. (2020), show that brominated VSLS have the largest impact





on $O_3$ destruction in spring, but the overall $O_3$ loss in the LMS is largest in winter. On the other hand, others suggested that the changing transport is the main driver for the variability and trends in LS $O_3$ (e.g., Chipperfield et al. (2018)).

Further, it is well known that the following factors relevant for stratospheric $Br^{tot}$ may vary strongly with location and
season: (a) the source strengths of VSLS and sea salt at the surface (e.g., Quack et al. (2004); Yokouchi et al. (2005); Warwick et al. (2006); Liang et al. (2010); Ordóñez et al. (2012); Ziska et al. (2013); Hossaini et al. (2012b, 2013, 2016); Wang et al. (2019), and many others), (b) the delivery of VSLS, their product gases and eventually of halogens sourced from sea salt to the stratospheric entry level (e.g., Aschmann et al. (2009, 2011); Aschmann and Sinnhuber (2013); Liang et al. (2014); Navarro et al. (2015); Hossaini et al. (2016); Schmidt et al. (2016); Fiehn et al. (2018); Filus et al. (2020); Adcock et al. (2021), and
others), (c) the efficacy of removal of the halogenated product gases from the upper troposphere and lower stratosphere (UTLS) (e.g., Sinnhuber and Folkins (2006); Aschmann et al. (2011); Aschmann and Sinnhuber (2013)), and (d) the transport of air into LMS via the two major pathways, i.e., by stratosphere–troposphere exchange (STE) at mid-latitudes and the isentropic transport of air from the lower TTL into the LMS (e.g., Holton et al. (1995); Hoor et al. (2005); Bönisch et al. (2009); Ploeger et al. (2015); Konopka et al. (2015); Inai et al. (2019), and others). Therefore, it cannot be assumed ad-hoc that the sum nor the
speciation of the source and product gases for each halogen type (here of bromine) are spatially and temporally constant in the air masses of each transport pathway and hence the whole LMS.

The transport of air masses associated with the Asian monsoon anticyclone has recently attracted interest since it delivers pollutants and eventually VSLS to the LMS (e.g., Levine et al. (2007); Gettelman et al. (2011); Vogel et al. (2014, 2016); Orbe et al. (2015); Müller et al. (2016); Rolf et al. (2018) and Wetzel et al. (2020)), but has not been studied extensively for bromine
(Adcock et al., 2021). Studies such as by Vogel et al. (2014) discuss the rapid ($\sim 5$ week) transport pathway of air masses from the south-eastern Asia boundary layer to the LMS over northern Europe in September 2012 (spatially and seasonally similar to the current study). While bromine transport to the TTL has been observed in various studies, the continued transport to the LMS in boreal summer is still insufficiently investigated. In addition, it has been speculated that the varying sources strengths, transport pathways, removal procceses, and in particular the contributions from (3) and (4) to the LMS $Br^{tot}$, may change with
future climate change (e.g., Hossaini et al. (2012a); Falk et al. (2017) and Ziska et al. (2017)). However, direct observations on this varying composition, amount of bromine and its consequence for LMS $O_3$ is still lacking.

The present study addresses the spatial variability of $Br^{org}$ and $Br_y^{inorg}$ in the NH UTLS during fall 2017 as part of the WISE (Wave-driven and ISentropic Exchange) campaign. Together with measurements of transport tracers (CO and $N_2O$) and air mass age tracer ($SF_6$), corroborated by Lagrangian chemistry transport model simulations, the major transport pathways
of brominated VSLS and their product gases into the extratropical LS (Ex–LS), i.e., by STE at mid-latitudes and isentropic transport from the tropical UT/TTL by planetary waves, are discussed. Our study of $Br^{tot}$ thus partly complements the recent study of Keber et al. (2020) on the contribution and origins of the different organic source gases to $Br^{tot}$ in the NH Ex–LS.

However, observational studies addressing $Br^{tot}$ around the UTLS may not directly relate to $Br^{tot}$ available in the middle stratosphere because of the variability and heterogeneity in the source strengths, the variable transformation and atmospheric
transport of the brominated VSLS, and the amount of bromine inferred by the product gases and sea salt. In addition to this variability, the accuracies of the individual techniques (though they have been greatly improved over time), and the suspected





small trend in stratospheric bromine from controlled substances ($-0.15 \pm 0.04$ ppt year$^{-1}$; Engel and Rigby et al. (2018)), makes it difficult to infer a trend in Br$^{\mathrm{tot}}$ from measurements in the UT/TTL and LMS. Therefore, a continuation of high-precision Br$^{\mathrm{tot}}$ measurements in aged and well-mixed air of the middle stratosphere is necessary in the coming decades, both
for monitoring the contribution of the controlled bromine substances as well as the expected climate-change-induced changing contributions of VSLS and inferred Br$_{\mathrm{y}}^{\mathrm{inorg}}$ to stratospheric Br$^{\mathrm{tot}}$.

The paper is organized as follows. The key instruments and methods along with some (necessary) data reductions are reported in Sect. 2. The WISE campaign measurements are described in Sect. 3. Inferred Br$^{\mathrm{tot}}$ measurements are presented in Sect. 4.1, followed by their geographical distribution in the UTLS indicating their source regions in Sect. 4.2. Model results of
transport pathways contributing to elevated Br$^{\mathrm{tot}}$ air masses in the LMS are discussed in Sect. 4.3, and the consequences of this additional Br$^{\mathrm{tot}}$ in the LMS on O$_3$ is discussed in Sect. 4.4. Finally, our conclusions are summarized in Sect. 5.

## 2 Instruments and Methods

The present study uses four instruments assembled on the German High Altitude and LOng range research aircraft (HALO). They are the mini-DOAS instrument for remote sensing measurements of O$_3$, NO$_2$, and BrO (Hüneke et al., 2017), FAIRO
(Fast Airborne Ozone monitor) for the measurements of in situ O$_3$ (Zahn et al., 2012), GhOST (Gas chromatograph for the Observational Studies using Tracers) for the in situ measurements of the organic brominated source gases (Sala et al., 2014) and SF$_6$ mean air mass lag-times (Bönisch et al., 2009), and UMAQS (University of Mainz Airborne QCL (quantum cascade laser) Spectrometer) for in situ measurements of air mass transport tracers N$_2$O and CO (Müller et al., 2015; Kunkel et al., 2019). For the interpretation of our measurements regarding the chemistry and the transport of air masses, we use simulations by
the Lagrangian chemistry transport model (CTM) CLaMS (Chemical Lagrangian Model of the Stratosphere) (e.g., McKenna et al. (2002b); Pommrich et al. (2014)). In addition, the O$_3$ loss caused by the transport of bromine-containing substances into the NH UTLS is simulated by the TOMCAT/SLIMCAT model (Chipperfield, 1999, 2006). Since all methods and tools have previously been described in detail in dedicated studies, only their major features are briefly summarized here.

### 2.1 Mini-DOAS Measurements of O$_2$, NO$_2$, and BrO

The mini-DOAS instrument, operated by the University of Heidelberg on the HALO aircraft, is a remote sensing instrument consisting of UV/vis/near–IR six–channel optical spectrometers. The scattered skylight received from scanning limb and nadir directions can be analyzed for O$_3$, NO$_2$, and BrO (as well as a variety of other trace gases). In this study only the UV and visible channels (FOV is $0.5°$ vertical, $3.15°$ horizontal) in the limb orientation are used. Therefore, the probed air masses are located perpendicular to the aircraft main axis on the starboard side and the telescopes are continuously adjusted in flight to
compensate for changes in the roll angle of the HALO aircraft for constant alignment with the flight altitude. Data collected from misaligned telescopes are excluded in the analysis. For extensive details of the HALO mini-DOAS instrument see Hüneke (2016) and Hüneke et al. (2017) and references therein. For additional mini-DOAS measurements see Werner et al. (2017) and Stutz et al. (2017) and for more details of the mini-DOAS measurements from HALO see Kluge et al. (2020).



The post-flight spectral analysis of the collected data for the detection of $O_3$, $NO_2$, and BrO is based on the DOAS method
(Differential Optical Absorption Spectroscopy (Platt and Stutz, 2008)). For the UV spectrometer retrieval of BrO differential
slant column denisities (dSCD), given in molecules $cm^{-2}$, a fit window of $346-359$ nm is used as suggested by Aliwell
et al. (2002), i.e., a two–band fit of the vibrational transitions 4,0 and 5,0 of the $A^2\Pi_{3/2} \leftarrow X^2\Pi_{3/2}$ electronic transition
(Wilmouth et al., 1999). For $O_3$–UV dSCDs, used as the corresponding scaling gas, a fit window of $338-357$ nm is applied
(Pukīte and Wagner, 2016). $NO_2$ dSCDs from the visible channel spectrometer are evaluated between $432-490$ nm (adjusted
lower limit compared to Hüneke et al. (2017) because the wavelength range of the updated spectrometer changed, therefore
the lower wavelengths are removed to avoid instabilities). And $O_3$–vis dSCDs are evaluated between $450-500$ nm (Hüneke
et al., 2017). A summary of the trace gases and details of their absorption cross sections used for the DOAS analysis can be
found in Tables 1 and 2. The dSCDs of each gas are calculated using an in–flight Fraunhofer reference spectrum, which is the
same for the scaling and target gas (e.g., $O_3$–UV and BrO, respectively). A running mean of five spectra (on average 158 s
per running mean spectrum and corresponds to $\sim 31.6$ km with a $1\sigma$ variability of 55 s) is used for BrO (and $O_3$–UV) dSCD
analysis to improve the signal to noise due to its low volume mixing ratios (VMRs), while for $NO_2$ (and $O_3$–vis) each spectrum
is evaluated individually (on average 12 s per spectrum corresponding to $\sim 2.4$ km with a $1\sigma$ variability of 5 s). Outliers of
selected individual dSCD data points that deviate by more than $5 \times 10^{13}$, $1 \times 10^{15}$ or $2 \times 10^{18}$ molecules $cm^{-2}$ ($\sim 10-25\%$
jump) from the nine–point running mean for BrO, $NO_2$, or $O_3$, respectively, are averaged to remove unstable spectra. Typical
dSCD errors of BrO, $NO_2$, and $O_3$–UV/vis can be found in the final column of Table 1. Additional settings for the DOAS
analysis are described in detail in Hüneke et al. (2017).

The summation of the dSCD and a reference slant column density ($SCD_{ref}$), which accounts for the absorption in the
Fraunhofer reference chosen from the flight, results in the total SCD (see Eq. (1) in Hüneke et al. (2017)). The $O_3$–UV $SCD_{ref}$
is found by evaluating the Fraunhofer reference using an ozone-free external high–resolution absorption–free Sun spectrum
from Chance and Kurucz (2010). The BrO $SCD_{ref}$ is calculated by using the quasi–linear relationship between BrO dSCDs
and the absorption-free spectra (Chance and Kurucz (2010)) SCDs of $O_3$–UV; any data points from the initial aircraft acent and
final descent as well as when the solar zenith angle (SZA) is $> 89°$ are removed. When inspecting the $O_3$ vs BrO relationship,
an offset of BrO when there is no $O_3$ present is identified and represents the unaccounted BrO absorption in the chosen
Fraunhofer reference and hence is the BrO $SCD_{ref}$. In the visible channel, the $NO_2$ and $O_3$–vis $SCD_{ref}$s are both simulated
using the Monte Carlo radiative transfer (RT) model (McArtim; Deutschmann et al. (2011)) because the inferred $O_3$–vis
SCDs in the chosen visible wavelenth range by the absorption free spectrum from Chance and Kurucz (2010) are frequently
unreliable.

For the retrieval of VMRs at flight altitude the recently developed $O_3$–scaling method is used, the details of which are
discussed in Stutz et al. (2017), Hüneke et al. (2017), and Werner et al. (2017). For this method, the radiative transfer of each
limb measurement is modelled using McArtim (Deutschmann et al., 2011) and considers the atmospheric, instrumental, and
celestial (e.g., Sun position) parameters and geolocation as described in Hüneke et al. (2017). In the RT model, a combination
of climatological profiles of the aerosol concentration are used, including EARLINET data (European Aerosol Research Lidar
Network; Pappalardo et al. (2014)) and SAGE III-ISS data (Stratospheric Aerosol and Gas Experiment III on the International





Space Station; NASA/LARC/SD/ASDC (2017)). The a priori profile shapes of the scaling and target gases are taken from
simulated trace gas VMR curtains along the flight track modelled by CLaMS (see Sect. 2.5 below). To remove any discrepancies
between the CLaMS model and the real conditions particularly when crossing a strong $O_3$ gradient, the CLaMS results for all
gases are vertically adjusted (up or down) to match the CLaMS $O_3$ at flight altitude with the in situ $O_3$ observed by FAIRO (see
Sect. 2.2 below). This was done previously in Werner et al. (2017) and Stutz et al. (2017), who shifted the TOMCAT/SLIMCAT
(see Sect. 2.6 below) model results to match the measured $O_3$. The vertical shifting of CLaMS is on average $\sim 1$ km which
corresponds to approximately one pressure/altitude grid box of the original CLaMS resolution and with a maximum shift of
2.5 km. These adjusted profiles are used by the RT model to calculate the trace gas absorption in the line-of-sight to the total
absorption, i.e., the so called $\alpha$ factors (see Stutz et al. (2017) or Hüneke et al. (2017) for a derivation).

Lastly, to retrieve the trace gas VMRs ($[X]_j$) in an atmospheric layer j (i.e., at flight altitude for limb measurements), the
scaling method equation is applied (see Stutz et al. (2017) or Hüneke et al. (2017)):

$$[X]_j = \frac{\alpha_{X_j}}{\alpha_{P_j}} \cdot \frac{SCD_X}{SCD_P} \cdot [P]_j \tag{1}$$

where X represents the target gas (BrO or $NO_2$ in this study) and P is the scaling gas ($O_3$ in this study). The scaling method
virtually compensates for any effects of the light path length modifications (e.g., scattering by aerosol and clouds) observed in
the SCDs and includes instrumental, celestial and climatalogical data mentioned above in the $\alpha$ factor. The $\alpha_{X_j}$ of the target
gas and $\alpha_{P_j}$ of the scaling gas are wavelength dependent, and therefore the target and scaling gas should have similar or largely
overlapping wavelength fit windows (Table 1). However, for a given wavelength the $\alpha$ factor ratio is insensitive to scattering
by aerosols and clouds due to the equivalence theorem (Irvine, 1964) as shown in simulations performed by Knecht (2015) or
in the supplement of Hüneke et al. (2017). The $SCD_X$ and $SCD_P$ are the total slant column densitites of the target gas (BrO
or $NO_2$) and scaling gas ($O_3$), resepectively, evaluated from the mini-DOAS data. $[P]_j$ is the scaling gas VMR, which in this
study is the in situ $O_3$ as measured by the FAIRO instrument aboard HALO (see Sect. 2.2). The VMR of the target gas, $[X]_j$,
for these limb UV/vis measurements is an average over a given atmospheric volume based on the distance the aircraft has
travelled during a single measurement, over the FOV of the telescope to the right of the aircraft, and the light path length in
the current conditions (for examples see Figs. 2 and 6 of Kluge et al. (2020)). Typical precision errors for the VMRs calculated
by the $O_3$–scaling method are $\pm 0.6$ ppt for BrO and $\pm 8$ ppt for $NO_2$. For a detailed description and a breakdown of the error
budget refer to Hüneke et al. (2017).

## 2.2 FAIRO $O_3$ Measurements

The Fast AIRborne Ozone monitor (FAIRO) is a fully custom-built, lightweight and compact $O_3$ sensor (14 kg, 19″, 13.5 cm
high) operated by Karlsruhe Institute of Technology. FAIRO combines (a) the absolute measurement by a two-channel UV
photometer detecting the absorption of $O_3$ in the Hartley band around 255 nm, and (b) the detection with a fast chemilumines-
cence sensor (CL; >10 Hz) (Zahn et al., 2012). The CL sensor detects light emission from the reaction of $O_3$ with an organic
dye, coumarin (e.g., Ermel et al. (2013)). The CL response is calibrated by the UV photometer, thus combining the accuracy
of a UV photometer with the high measurement frequency of a CL detector. Typical $1\sigma$ measurement precision is 0.08 ppb for





the UV-spectrometer measuring at 0.25 Hz and 0.5% for the CL detector measuring at 10 Hz. Accuracy for the UV spectrometer measurements is $\sim 1.5\%$ (mostly determined by the uncertainty of the $O_3$ cross section). Throughout this study, the $O_3$ from the CL method is used with the exception of one flight (on 9 October 2017 – RF11, refer to Sect. 3) for which the UV
photometer method $O_3$ is used as only partial CL $O_3$ data was available.

## 2.3  GhOST Measurements of Brominated Source Gases, Transport Tracer $SF_6$ and Air Mass Lag-Time

The Gas chromatograph for the Observational Studies using Tracers (GhOST) instrument, operated by the University of Frankfurt, detects stable atmospheric trace gases. It has two gas chromatographic channels, one which is coupled to a mass spectrometer (MS; Agilent MSD 5975) and the other one to an electron capture detector (ECD) (e.g., Sala et al. (2014), Obersteiner
et al. (2016), Keber et al. (2020)). $SF_6$ and CFC-12 (chlorofluorocarbon) are measured on the ECD channel with a time resolution of 60 s and a high precision of typically about 0.6% for $SF_6$ and 0.2% for CFC-12, in a similar setup as used in Bönisch et al. (2009). The mean age of air can be inferred from $SF_6$ measurements only if a single entry point can be assumed. As air masses observed during this study, i.e., the WISE campaign, are strongly influenced by transport of air from the NH mid latitudes and tropics, a direct derivation of mean age is not possible. Instead we use so-called lag-time here, i.e., the time lag
between the occurrence of a certain $SF_6$ VMR in the UTLS with respect to the time when this mixing ratio was observed in the global mean. In addition, $SF_6$ data are smoothed using a local correlation with the CFC-12 data (an atmospheric transport tracer) which has a higher precision. Ten data points surrounding the measurement on both sides are used to determine the local correlation for each data point. This procedure retains the local information without introducing an offset to $SF_6$ mixing ratios and removes some instrumental scatter. The lag-time derived in this way has an overall precision better than 0.3 years and an
estimated accuracy of 0.6 years. The global mean $SF_6$ VMR used as a reference is lower than the NH VMRs. In particular close to the TP, observed mixing ratios are often higher than those in the reference time series, resulting in negative lag-times.

The second channel measures bromocarbons and other halogenated species with cryogenic enrichment (Obersteiner et al., 2016), gas chromatographic separation, and mass spectrometric detection with a time resolution of four minutes (Sala et al., 2014). Atmospheric mixing ratios of $CH_3Br$, halons H–1301, H–1211, and H–2402 as well as the short-lived bromocarbons
($CH_2Br_2$ and $CHBr_3$) are reported on the scales used in the AGAGE network (Prinn et al., 2018). Mixed bromochlorocarbons ($CH_2BrCl$, $CHBr_2Cl$, and $CHBrCl_2$) and H–1202 are calibrated based on intercalibration experiments with the UEA. Due to a decrease of $CH_3Br$ in the calibration gas and poor signal to noise ratios for this compound, a correlation between H–1211 and $CH_3Br$ measured during the PGS (three part mission: POLSTRACC (Polar Stratosphere in a Changing Climate), GW-LCYCLE (Investigation of the Life cycle of gravity waves) and SALSA (Seasonality of Air mass transport and origin
in the Lowermost Stratosphere)) campaign has been used. This correlation leads to a scaling of 12.5% so as to arrive at typical tropospheric mixing ratios of $CH_3Br$ observed in the NH in 2017 and then applied to the data of the current study. Typical detection limits for the different substances during WISE are given in Keber et al. (2020). The $1\sigma$ scatter (the standard deviation) of $Br^{org}$ measurements in a 10 K potential temperature interval directly below the TP is $\pm 0.81$ ppt (4%), which includes the precision of the measurement as well as the atmospheric variability (Sala et al., 2014).





## 2.4 UMAQS Measurements of $N_2O$ and CO Transport Tracers


The University of Mainz Airborne QCL (quantum cascade laser) Spectrometer (UMAQS) is operated by the University of Mainz. It simultaneously measures nitrous oxide ($N_2O$) and carbon monoxide (CO) from the HALO aircraft. The instrument is in situ calibrated against secondary standards, which are compared to NOAA primary standards before and after the campaign. It uses direct absorption of infrared radiation at $2200\ cm^{-1}$ using a Hariott cell with a $76\ m$ light path at $50\ hPa$. Measurements

are recorded with a frequency of $1\ Hz$, and the total uncertainty during this study was determined to be $0.18\ ppb$ for $N_2O$ and $0.9\ ppb$ for CO. These tracers are used for the interpretation of air mass transport around the UTLS during this study. For more details refer to Müller et al. (2015) and Kunkel et al. (2019).

## 2.5 Lagrangian Modelling by CLaMS

The Chemical Lagrangian Model of the Stratosphere (CLaMS) is developed at Forschungszentrum Jülich in Germany. It is

a Lagrangian CTM (McKenna et al. (2002a, b); Pommrich et al. (2014) and references therein), which utilizes wind and temperature data of the ECMWF (European Centre for Medium-Range Weather Forecasts) from ERA-Interim re-analysis data (Dee et al., 2011). CLaMS is particularly good at portraying the gradients of trace gases and the transport of air parcels especially in the region of transport barriers for example near the extratropical tropopause (Ex–TP) due to its Lagrangian nature (e.g. Vogel et al. (2011, 2016)). Aura-MLS (Microwave Limb Sounder) and ACE-FTS (Atmospheric Chemistry Experiment -

Fourier Transform Spectrometer) satellite data as well as tracer-tracer-correlations are used to initialize CLaMS simulations. The simulation used here starts on 1 May 2017 and has a horizontal resolution of $100\ km$.

The bromine chemistry in the simulations is setup to optimally reproduce reactions in the stratosphere but lacks accuracy for photochemical processes in the troposphere as the heterogeneous reactions of brominated species on ice and liquid water are not yet implemented (Grooß et al., 2014). Also, only long-lived $Br_y^{inorg}$ is part of the simulation and bromine compounds

from VSLS are considered to already be converted to $Br_y^{inorg}$. This may cause offsets in $Br_y^{inorg}$, but nevertheless ratios like $BrO/Br_y^{inorg}$ are reasonally determined by this method.

Trace gas curtains along the flight tracks have been derived for each flight of the WISE campaign. These curtains are defined by the latitude, longitude, and time of the flight track for many altitudes. CLaMS model information for these points was achieved using back-trajectories to the model output time of the previous day at 12:00 UTC. The chemical composition from

the 3-D CLaMS simulation is then interpolated to the trajectory endpoints, subsequently the CLaMS chemistry is calculated forward in time along these trajectories to determine the chemical composition along the given curtain. The CLaMS curtains has a one min time resolution (corresponding to $\sim 13\ km$) and 41 pressure levels between $10-1000\ hPa$. For the present study, CLaMS curtains are used as input for the RT simulations for the mini-DOAS data analysis (see Sect. 2.1). These simulated trace gas mixing ratios along the flight track are used for direct comparisons with the measured data of $O_3$, $NO_2$, and $BrO$ and

the simulated $BrO/Br_y^{inorg}$ partitioning coefficient is used to infer measured $Br_y^{inorg}$ from mini-DOAS $BrO$ data.

In previous studies, different kinds of artificial tracers have been implemented in CLaMS to identify the transport pathways and to quantify the transport of air masses from defined domains/regions to other locations of the atmosphere considering



advection and mixing (e.g., Vogel et al. (2011, 2019)). To compare the WISE measurements with the CLaMS artificial tracers, the simulations are interpolated in space and time to the position of the HALO flight track using CLaMS trajectories (started every second along the flight track). In this study we use two different kinds of artificial tracers. The first kind is an artificial air mass origin tracer initialized on 1 May 2017 approximately 5 months prior to the WISE campaign. The air mass origin tracers mark nine distince 3-D domains in the entire model atmosphere which are the tropical troposphere, upper and lower TTL, tropical pipe, mid-latitudinal troposphere and LMS, polar troposphere and LMS and lastly the stratospheric background. These are used to study vertical and horizontal transport of these air masses to the LMS in the NH within 5 months. Divisions of the air mass origin domains are described in further detail in Table 1 and Fig. 4 of Vogel et al. (2011).

The second kind is an artificial tracer that marks specific regions in the CLaMS model boundary layer referred to as surface emission tracers and are used here to study the transport of young air masses into the LMS in the NH. In contrast to the air mass origin tracers discussed above which are initialized once on 1 May 2017, the different surface emission tracers are continuously released (every 24 h) in the model boundary layer (2−3 km above the surface following orography) from the start of the simulation on 1 May 2017 (for more details see Vogel et al. (2015, 2019)). The surface emission tracers are divided into 24 surface regions (see Fig. 9 below) defining relatively small regions in south-eastern Asia to study in particular the influence of air masses from the Asian monsoon and its tropical adjacent regions on transport into the LMS. Moreover, in this study, the entire Earth's surface is covered by surface emission tracers. For some regions, coastlines or orography (both provided by ECMWF) are used as criteria to define the boundaries between different regions. The same set-up for the surface emission tracers was already used for WISE measurements in Wetzel et al. (2020).

## 2.6 Chemical Transport Modelling by TOMCAT/SLIMCAT

The TOMCAT/SLIMCAT (hereafter TOMCAT) 3-D off-line global CTM (Chipperfield, 1999, 2006) is used here for the interpretation of the transport of trace gases from the tropical UT/TTL and Ex–TP regions into the LMS and to assess the corresponding loss in $O_3$. Similarly to CLaMS, the off-line model is driven by meteorology (e.g., horizontal winds, temperature, surface pressure, and convective mass fluxes) from ECMWF ERA-Interim reanalyses (Dee et al., 2011), with no feedback of the chemistry on the dynamics. In the setup used here the model employs a hybrid vertical coordinate with terrain-following sigma levels near the surface and pressure levels at higher altitudes. Large-scale vertical motion is calculated from the divergence of the horizontal winds. In our simulations the model uses a detailed stratospheric chemistry scheme TOMCAT (e.g. see Chipperfield et al. (2018)) with photochemical information from JPL-2015 (Sander et al., 2011). The $O_3$ loss reactions are described in Werner et al. (2017) Appendix A.

Initial constraints for the brominated species in the UT are given in Table 3. We have performed three different TOMCAT sensitivity runs from January 2016: a Base run followed by two runs with elevated bromine constraints. The Base run initializes the model with constant bromine contributions along the entire TP with the Ex–TP boundary conditions with $[Br^{tot}]$=19.6 ppt. The additional two runs continue to use the Ex–TP conditions in the extratropics, while elevated bromine in the form of $Br^{VSLS}$ and $Br_y^{inorg}$ are initialized in the tropical UT for two differing scenarios. Run 1 uses $Br^{VSLS}$ constraints from Navarro et al. (2015) and $Br_y^{inorg}$ from Werner et al. (2017) resulting in $[Br^{tot}]$=20.8 ppt for the tropical UT. Run 2 uses $Br^{VSLS}$ and





$Br_y^{inorg}$ VMRs from the current study in the tropical UT/TTL resulting in $[Br^{tot}]$=21.7 ppt. These runs, with differing bromine VMRs in the tropical UT but with constant Ex–TP constraints, simulate different air masses being transported into the LMS. Comparison of the model runs allows the diagnosis of the impact of the differing bromine levels on the $O_3$ levels in this region.

## 3 Measurements: WISE Campaign

For the present study, we analyze simultaneous measurements of bromine collected during 15 flights on the German HALO research aircraft during the WISE campaign based from Shannon, Ireland. Specific topics of interest during WISE include the transport pathways, time-scales and mixing processes affecting the late summer/fall UTLS. A particular focus was on the role of the transport of air from the TTL and Asian monsoon region on $H_2O$, trace gas budgets including VSLS and their product gases in the LMS, as well as small scale processes leading to irreversible exchange across the Ex–TP by Rossby wave-breaking events (Homeyer and Bowman, 2013).

The research flights in this study, RF02–RF16, were conducted from 13 September 2017 until 21 October 2017 with the majority of the $\sim 140$ flight hours probing the extratropical UTLS in mid-to-high latitudes over the North Atlantic, Norwegian Sea and north-western Europe. The flight tracks are shown in Fig. 1 with a few research flights emphasized by dashed lines, to be discussed in more detail below. The flights were mainly during daytime, but three take offs and six landings were near sunrise/sunset or in the dark. The trace gas measurements of $O_3$, $N_2O$, CO, and $SF_6$ are available for all research flights. $NO_2$, BrO, and $Br_y^{inorg}$ VMRs are also available for all flights, however only at SZA $< 89°$. $Br^{org}$ VMRs are not available during four flights (RF07, RF09, RF14, and RF15), and during three other research flights (RF02, RF06, and RF11) $Br^{org}$ VMRs are only available for a portion of the flight time. As such, inferred $Br^{tot}$ measurements are available for 11 of the 15 WISE research flights.

The measurement region ranged from longitudes $48.4°$ W $- 15.7°$ E and latitudes from $37.9°$ N $- 74.4°$ N. In equivalent latitudes (Butchart and Remsberg, 1986), the investigated air masses ranged from those being recently transported from the tropics ($\sim 6°$ S) up to $86°$ N. The majority of the flight time was spent at altitudes between $8.5-14.5$ km allowing for measurements in the UTLS and crossing the TP frequently. The maximum potential temperature reached was $\Theta$=410 K, allowing for observations throughout the LS.

Two flights are chosen as examples. One has typical $Br^{tot}$ mixing ratios near the campaign LS averge along with a few segments observing elevated $Br^{tot}$ in the LMS (RF04; Fig. 1 blue dashed line). The other has elevated $Br^{tot}$ from former tropical UT air masses (RF08; Fig. 1 yellow dashed line).

The first sample flight, RF04 on 20 September 2017, consisted of two different length, but overlapping, rectangular shuttles northeast of Ireland up the Scandinavian coast, over the Norwegian Sea and only briefly crossing landmasses over the United Kindom and Ireland (see Fig. 1 and 2). The flight track reached a maximum latitude of $69.4°$ N and longitude of $11.2°$ E. The final stretch back to Shannon was flown at the highest altitude during this flight of $14.35$ km and reaching a maximum potential temperature of $\Theta$=399 K, corresponding to $\Delta\Theta$=71 K above the WMO (World Meteorological Organization) TP. The BAHAMAS (BAsic HALO Measurement And Sensor system) flight level (black solid line) and WMO TP (black dashed





line) are shown in potential temperature in Kelvin along with the equivalent latitude (blue, right axis) in Fig. 2 (a). The TP was crossed a few times, but the majority of the flight time was spent in the LMS. The mean lag-time of the air masses ranged from $-1.08$ to $1.85$ years (Fig. 2 (f)). The negative mean air mass lag-times are due to higher $SF_6$ VMRs near the NH source regions (compared to the global mean) and in particular in the upper levels of the tropical TP region. This flight is chosen as a sample flight of $Br^{tot}$ mixing ratios near the LS campaign average as well as flight segments within the LMS region with

elevated bromine indicated by the dark gray shading discussed in Sect. 4.2.

The second sample flight, RF08 on 1 October 2017, probed the remnants of Hurricane Maria. This flight consisted of two overlapping rectangular shuttles southwest of Ireland over the Celtic Sea and North Atlantic Ocean (see Figs. 1 and 3). The latitudes ranged from $45.8°-63.0°$ N, while the longitude of the rectangular patterns ranged between $15.0°$ W $- 8.6°$ W. The highest altitude during this flight was $14.36$ km. The maximum potential temperature reached was $\Theta=398$ K, which also

corresponds to the furthest distance above the WMO TP with $\Delta\Theta=84$ K. As during all WISE flights, the TP was crossed several times shown in Fig. 3 (a), with the mean lag-time ranging from $-1.10$ to $1.96$ years (Fig. 3 (f)). Large sections of this flight observed subtropical as well as former tropical UT/TTL air masses. These air masses transported from the tropics with high potential temperature ($\Theta > 355$ K) and low potential vorticity units ($< 2$ PVU) are furtheron called tropical $\Theta_{high}/PV_{low}$ air, indicated by the light gray shading. This flight is chosen representing the variety of air masses observed during WISE,

resulting in elevated $Br^{tot}$ mixing ratios largely due to the more abundant VSLS in the tropical $\Theta_{high}/PV_{low}$ air.

Figures 2 and 3 provide overviews of the $O_3$ measured by FAIRO, CO measured by UMAQS, the organic bromines measured by GhOST-MS, and $NO_2$ and BrO measured by mini-DOAS. The modelled $O_3$, $NO_2$, and BrO from the CLaMS collectively vertically-shifted and thereby adjusted to match in situ measured $O_3$ (refer to Sect. 2.1) curtains shows excellent agreement to measurements particularly for the former two gases. Additionally, the CLaMS calculated ratios of $BrO/Br_y^{inorg}$

and $BrONO_2/Br_y^{inorg}$ are displayed, as well as CLaMS artifical surface emission tracers from the tropical marine environment (TME) in south-eastern Asia and Central America (CAM) (discussed in Sect. 4.3). The variety of air masses observed during this campaign is not only reflected in $Br^{tot}$ fluctuations but also in the range of air mass lag-times, which vary between $-1.05$ to $2.33$ years.

Complementary measurements and insight for a suite of sources gases and studies of the dynamics, potential vorticity,

baroclinic waves and isentropic mixing of the region have been reported elsewhere for flights during the WISE campaign (e.g., Fix et al. (2019); Kaluza et al. (2019); Kunkel et al. (2019); Hauck et al. (2020); Keber et al. (2020); Wetzel et al. (2020)). More studies from the WISE campaign addressing wave-driven isentropic exchange in the NH UTLS are published in the ACP/AMT/WCD inter-journal special issue (see https://www.atmos-chem-phys.net/special_issue1061.html).

## 4 Results and Discussion

### 4.1 Total Bromine as a Function of $\Theta$-Distance from the Tropopause

Here we report on the UTLS inferred total bromine, $Br^{tot}$, for all available data during the WISE campaign in the fall of 2017. Figure 4 shows the additive break down of the organic ($CH_3Br$ – dark blue, halons – purple, and $Br^{VSLS}$ – light blue) and





inorganic (BrO measurements scaled by the modelled partitioning coefficient $Br_y^{inorg}/BrO$ – red points) bromine components with the potential temperature distance from the WMO TP in Kelvin, defined as $\Delta\Theta = \Theta - \Theta_{Tropopause}$. Data points where the SZA > 89° may introduce additional photochemistry-related uncertainties in the $Br_y^{inorg}$ measurements (e.g., Kreycy et al. (2013)) and therefore are excluded from the following discussion. The weighted mean $Br^{tot}$ in the UTLS (all measurements) is $19.4 \pm 1.3$ ppt (the uncertainty being the $1\sigma$ variability of the measurements). Excluding the tropospheric measurements, the LS has a weighted average of $[Br^{tot}] = 19.2 \pm 1.2$ ppt, designated by the black solid line in Fig. 4. A more rigorous criteria with which to exclude data such as SZA > 85°, as well as excluding data based on the modelled partitioning coefficient $BrO/Br_y^{inorg}$ < 55%, only impacts the $Br^{tot}$ weighted mean by $\sim 0.1$ ppt decrease. $Br^{tot}$ ranges from $16.2-23.4$ ppt (with the exception of two larger outliers: one from large VSLS VMRs near the surface and another at approximately 35 K above the TP because of an instability of the $Br_y^{inorg}$ retrieval). The variability is partly due to the different origins of the air masses and various inferred mean air mass lag-times which have a spread of more than three years. Inferred $Br^{tot}$ measurements are up to $\Delta\Theta=88$ K above the WMO TP, with the measurements mostly in the LMS during the WISE campaign.

The LS mean of $[Br^{tot}] = 19.2 \pm 1.2$ ppt within the given uncertainties is in good agreement with previous data (Engel and Rigby et al. (2018)). The expected $[Br^{tot}] = 19.6$ ppt for 2016 is primarily based on balloon soundings in the middle stratosphere and the expected trend in stratospheric bromine since the last balloon soundings in 2011. The trend is suspected to now be decreasing by $-0.15 \pm 0.04$ ppt Br yr$^{-1}$ between 2012 and 2016 (see Fig. 1-15 in Engel and Rigby et al. (2018)). Furthermore, our results fit well into the expectation based on the total amount of stratospheric $Br^{tot}$, published earlier by our group from different measurements (e.g., Dorf et al. (2006a) and Werner et al. (2017)) and others (e.g., Hendrick et al. (2008)).

Within $\Delta\Theta = \pm 5$ K of the TP, the weighted average $Br^{org}$ (light blue points in Fig. 4) is $17.8 \pm 1.2$ ppt, with several ppt larger bromine mixing ratios below the TP. Further into the LS ($\Delta\Theta=78-88$ K above the TP), $Br^{org}$ quickly decreases to $13.5 \pm 1.2$ ppt as $Br^{org}$ is being converted to $Br_y^{inorg}$ species. $CH_3Br$ (contribution (1) to $Br^{tot}$) is destroyed with increasing distance into the stratosphere at a slow rate from $7.1 \pm 0.2$ ppt within $\Delta\Theta = \pm 5$ K to $6.1 \pm 0.5$ ppt measured within $\Delta\Theta=78-88$ K. The tropospheric $CH_3Br$ ($7.3 \pm 0.2$ ppt) is slightly larger than the 2016 global annually averaged 6.8 ppt as reported in Engel and Rigby et al. (2018), but agrees well with the NH measurements of $\sim 6.75-7.75$ ppt from NOAA measurements which show that NH $CH_3Br$ is always larger than in the southern hemisphere (SH) (Fig. 1-7 of Engel and Rigby et al. (2018)). Contribution (2) to $Br^{tot}$, the four different halons (H–1301, H–1211, H–2402, and H–1202), has a mean VMR of $7.5 \pm 0.2$ ppt within $\Delta\Theta = \pm 5$ K of the TP (in good agreement with the annually averaged tropospheric 7.8 ppt as from 2016 reported by WMO (2018)) and also decreases slowly to $6.5 \pm 0.5$ ppt within $\Delta\Theta=78-88$ K in the LS. The larger part of the $Br^{org}$ decrease in the LS is by contribution (3), the VSLS ($CH_2Br_2$, $CHBr_3$, $CH_2BrCl$, $CHBrCl_2$, and $CHBr_2Cl$), due to their short global lifetime of less than 6 months. Within $\Delta\Theta = \pm 5$ K, the $Br^{VSLS}$ is $3.1 \pm 0.9$ ppt (up to 7.2 ppt) while at $\Delta\Theta=78-88$ K above the TP the $Br^{VSLS}$ decreases to $1.0 \pm 0.3$ ppt. As such, the accumulated decrease of $Br^{org}$ (contributions (1)–(3) to $Br^{tot}$) throughout the LS between the TP and $\Delta\Theta=88$ K above the TP is $\sim 4.3$ ppt.

Although further into the stratosphere the $Br^{org}$ is destroyed, bromine is converted to $Br_y^{inorg}$ and is compensated for in the $Br^{tot}$ budget. Contribution (4) to $Br^{tot}$, the $Br_y^{inorg}$, near the TP ($\Delta\Theta = \pm 5$ K) is around $1.5 \pm 0.6$ ppt (with a range of $0.2-3.3$ ppt) and increases to $5.8 \pm 1.8$ ppt ($2.9-10.1$ ppt) within $\Delta\Theta=78-88$ K above the TP. The conversion of $\sim 4.3$ ppt





$Br^{org}$ to $Br_y^{inorg}$ is evident from the $Br^{tot}$ budget, which is approximately constant for the entirety of the air masses probed during the WISE campaign. In the LS the weighted mean $Br^{tot}$ ranges from $18.7 \pm 1.2$ ppt (in the $\Delta\Theta{=}5{-}15$ K layer) to

$19.9 \pm 1.2$ ppt (in the $\Delta\Theta{=}35{-}45$ K layer). These small changes from the LS $Br^{tot}$ weighted mean of $19.2 \pm 1.2$ ppt do not result in an obvious trend between the WMO TP and $\Delta\Theta{=}88$ K above it but rather hint at air masses of differing $Br^{tot}$, which is discussed in the next sections.

## 4.2    Total Bromine as a Function of Equivalent Latitude Distribution in the NH

To identify potential source regions and transport pathways of $Br^{tot}$ into the LS, the various gases measured during WISE are

displayed in Fig. 5 with respect to potential temperature and equivalent latitude (Butchart and Remsberg, 1986). The VMRs of each trace gas is averaged horizontally in $5°$ equivalent latitude and vertically in 5 K grids to remove short term atmospheric variability. The campaign mean WMO TP varies from $\Theta \sim 316{-}374$ K at high to low equivalent latitudes, respectively.

The NH distributions of $Br^{org}$, $Br_y^{inorg}$, and $Br^{tot}$ are shown in Fig. 5 (a–c), respectively. Only 11 flights have $Br^{org}$ measurements available, resulting in a sparser distribution. Near the Ex–TP $Br^{org}$ mixing ratios are $\sim 18$ ppt, while in tropical

$\Theta_{high}/PV_{low}$ air $Br^{org}$ has higher mean VMRs of up to $20.5$ ppt, in agreement with previous studies (e.g., Navarro et al. (2015); Werner et al. (2017); Wales et al. (2018)). The maximum mean $Br_y^{inorg}$ VMR is $8.3$ ppt at $\Theta \sim 400$ K and $\sim 60°$ N equivalent latitude, in constrast to the $Br^{org}$ behavior. In the troposphere, $Br_y^{inorg}$ ranges from $\sim 0{-}3$ ppt. These results agree well with previous studies, such as Werner et al. (2017) who reported $[Br_y^{inorg}]{=}7.66 \pm 2.95$ ppt between $390{-}400$ K in the subtropical LS and reaching down to $0.5$ ppt in the tropical troposphere, as well as others (e.g., Wang et al. (2015); Koenig

et al. (2017)). A modelling study by Schmidt et al. (2016) reports $Br_y^{inorg}$ mixing ratios at the lower boundary of the TTL and Ex–TP of $3{-}4$ ppt, and daytime BrO mixing ratios of $1{-}2$ ppt, respectively.

Older air (see Fig. 5 (d)) has generally more $Br_y^{inorg}$ and less $Br^{org}$ present in large part due to the short lifetime of the VSLS. Averaged mean air mass lag-times reach above 2 years around $\sim 400$ K between $40{-}80°$ N equivalent latitudes and largely coincide with high $Br_y^{inorg}$ VMRs, with the exception of directly above the TP at $\sim 85°$ N equivalent latitudes. There, a small

pocket of older air (with lag-times larger than one year) has low $Br^{org}$ and $Br_y^{inorg}$. Within $\pm 5$ K of the TP, the mean lag-time is $-0.16 \pm 0.37$ yrs, with negative averaged lag-times down to $-0.5$ years in the troposphere. Here, the relative change of the mean air mass lag-times greater than 3 years is more relevant in the context of our study.

The average $Br^{tot}$ mainly varies between $18{-}20.5$ ppt both in the troposphere and stratosphere (see Fig. 5 (c)), however there are noticeable regions with higher bromine. Of particular interest is the region in the LMS at mid to high equivalent

latitudes where $Br^{tot}$ local grid means of up to $21.5$ ppt are observed. For the following discussion, this high bromine region (HBrR) is defined between potential temperatures of $\Theta{=}350{-}385$ K and equivalent latitudes of $55{-}80°$ N (designated by the black boxes in Fig. 5). Twelve out of the 15 research flights include flight segments within the HBrR, varying from minutes to several hours, with $Br^{tot}$ measurements available during eight flights (RF03–05, RF08, RF11–13 and RF16) over 8 hours. A frequency distribution of inferred $Br^{tot}$ is displayed in Fig. 6 (a) showing $Br^{tot}$ in the UTLS (blue) or only in

the LS (brown). The distribution of the $Br^{tot}$ in the HBrR (brown) as compared to the rest of the LS excluding the HBrR (blue) shows the characteristically significant increase of bromine in this region (Fig. 6 (b)). The HBrR has a weighted mean





[Br$^{\text{tot}}$]=19.9 $\pm$ 1.2 ppt. The HBrR measurements result in the bromine bulge noticeable in Fig. 4 around $\Delta\Theta = \sim 35$ K and are depicted by the right-pointing (gray and black-edged) triangles. Inferred Br$^{\text{tot}}$ measurements within the HBrR that are larger than the LS weighted mean by at least $1\sigma$ are designated by HBrR* (i.e., only the black-edged right-pointing triangles

in Fig. 4 and thicker black outlined bars in Fig. 6 (b)). The HBrR* has a Br$^{\text{tot}}$-weighted mean of $20.9 \pm 0.8$ ppt. Here we note that for individual measurements the estimated Br$^{\text{tot}}$ in the HBrR air is compatible with the LS average when considering the $1\sigma$ variability of the measurements. However, the clustering of the independently measured data, in particular of the HBrR* sample, suggests that the coadded error from the organic and inorganic measurements is actually lower than the stated $1\sigma$ variability of $\pm 1.2$ ppt, and thus the HBrR is a real and persistent feature over the course of the WISE campaign. HBrR* then

represents the elevated bromine transported to the LMS which is mixed with air of lower bromine content from the rest of the LMS and stratospheric background. The question is therefore what are the causes of this HBrR* air?

     Potential sources include other regions of elevated bromine larger than the LS average, such as from the tropical and extratropical troposphere. These regions have younger (tropospheric) air masses which correspond well with the relatively younger air masses observed within the HBrR (local grid mean lag-times ranging from $0.4-1.2$ years) compared to the surrounding

air (see Fig. 5 (d)). Elevated bromine in tropical $\Theta_{\text{high}}$/PV$_{\text{low}}$ air is observed with a weighted mean [Br$^{\text{tot}}$]=$21.6 \pm 0.7$ ppt, largely in agreement with previous experimental and modelling studies (Wang et al., 2015; Schmidt et al., 2016; Koenig et al., 2017; Werner et al., 2017). The bromine measurements in tropical $\Theta_{\text{high}}$/PV$_{\text{low}}$ air are from one flight, RF08 (see Fig. 3), over the course of almost 4 hours. One other flight (RF14) observed tropical $\Theta_{\text{high}}$/PV$_{\text{low}}$ air, although Br$^{\text{tot}}$ data is missing. Along with the elevated bromine in the tropical $\Theta_{\text{high}}$/PV$_{\text{low}}$ air mass, measurements of the transport tracers (Fig. 5 panels (e–h))

suggest that these air masses were eventually entrained by isentropic transport from the tropical $\Theta_{\text{high}}$/PV$_{\text{low}}$ region via the shallow branch of the Brewer-Dobson circulation (Levine et al., 2007; Birner and Bönisch, 2011). Near the Ex–TP ($35-55°$ N and $\pm 5$ K from the TP), Br$^{\text{tot}}$ has a weighted mean of $19.3 \pm 0.9$ ppt, indicating a possible second pathway of the air masses entraining bromine into the LMS by STE across the Ex–TP (e.g., Holton et al. (1995); Homeyer and Bowman (2013) and Kunkel et al. (2019)). A schematic summarizing these regions of elevated bromine and the transport pathways from the UT

into the LMS is shown in Fig. 7. Both possible transport pathways are investigated in the following with observational data from other measured gases (Fig. 5) and in Sect. 4.3 by transport modelling. The observational findings are as follows:

     (a) Elevated Br$^{\text{tot}}$ in tropical $\Theta_{\text{high}}$/PV$_{\text{low}}$ air masses has also been previously observed over the Pacific in 2013/2014, where inferred Br$^{\text{tot}}$ was measured to be up to 2 ppt larger in the TTL than expected for the middle stratosphere (Navarro et al., 2015; Werner et al., 2017; Wales et al., 2018). Studies such as by Fueglistaler et al. (2004); Levine et al. (2007); Gettelman et al.

(2011) and Ploeger et al. (2013) discuss the transport of air masses into the LMS that primarily stem from the tropics, in our study initiated by transport of the Asian monsoon (see Sect. 4.3). By the time tropical $\Theta_{\text{high}}$/PV$_{\text{low}}$ air reaches the LMS in the mid–high latitudes with transport times of $\sim 0.5-1.5$ years, significant portions of Br$^{\text{org}}$ have converted to Br$_y^{\text{inorg}}$.

     (b) Mean O$_3$ mixing ratios from FAIRO range from $\sim 100-760$ ppb in the UTLS (see Fig. 5 (e)). In the LMS, the O$_3$ and Br$_y^{\text{inorg}}$ VMRs often have a positive correlation, however it does not include the HBrR. In this region, higher Br$_y^{\text{inorg}}$ (and

Br$^{\text{org}}$) contributes to elevated Br$^{\text{tot}}$, while O$_3$ has a local grid mean minimum of $\sim 229$ ppb. Moreover, below this region





around $\Theta=330-350$ K and towards higher equivalent latitudes up to $90°$ N, there is an increase again of $[O_3]=250-340$ ppb while the $Br^{tot}$ decreases (the same trends are seen above the HBrR).

(c) $N_2O$ is a good transport tracer due to its long lifetime of 114 years (e.g., Ko et al. (2013)). The well-mixed tropospheric air with high $N_2O$ VMRs (measured by UMAQS) from the tropical $\Theta_{high}/PV_{low}$ region penetrates into the LMS above the TP extending to $350-380$ K and $50-80°$ N (Fig. 5 (g)), which coincides with the HBrR. Lower $N_2O$ VMRs are seen above and below this region. In the extratropics just above the TP, $N_2O$ shows an increasing tropospheric contribution from high to low equivalent latitudes. At equivalent latitudes $< 40°$ N, $N_2O$ shows almost tropospheric mixing ratios of $[N_2O] > 328$ ppb up to $\Theta=390$ K. Von Hobe et al. (2021) also observed elevated $N_2O$ mixing ratios in the tropics associated with the Asian summer monsoon in 2016/2017, extending up to $\sim 400$ K without sharp discontinuities at the lapse rate TP ($\sim 380$ K) or cold point TP ($\sim 390$ K). This indicates that the tropospheric air masses transported there are not yet mixed with stratospheric air masses. Von Hobe et al. (2021) further concluded that the main outflow of the 'fast convective chimney' from the Asian monsoon anticylconic region in the tropics was below 370 K with slower ascent rates above this as well as a smaller outflow from the chimney above 380 K, i.e., in direct agreement with our results of Fig. 5 (g).

(d) The $SF_6$ measurements are also used for analysis of the transport as it constantly increases and its atmospheric lifetime is in excess of 850 years (e.g., Kovács et al. (2017) and Ray et al. (2017)). The dominant transport from the tropical UT/TTL into the LMS has been previously investigated with $SF_6$ and $CO_2$ measurements (Bönisch et al., 2009). As seen in Fig. 5 (h), $SF_6$ VMRs range from $\sim 9.6$ ppt below the TP to $\sim 8.8$ ppt in the LMS. Similarly to $N_2O$, higher $SF_6$ VMRs (local grid means up to 9.3 ppt) protrude into the LMS at the tropical TP extending to $355-385$ K and $50-80°$ N which overlaps with the HBrR.

(e) At the same time, CO (from UMAQS shown in Fig. 5 (f)) with a shorter lifteime of $2-3$ months (Novelli et al., 1998) has an average of $[CO]=37 \pm 5$ ppb in the HBrR. The surrounding LS below and above the HBrR (i.e., from the TP to $\Theta=350$ K as well as $\Theta > 385$ K and between equivalent latitudes $55-80°$ N) has an average of $36 \pm 12$ ppb (influenced by higher CO near the TP) and VMRs reaching down to 22 ppb. The CO mixing ratios seen in the HBrR are considerably lower than in the tropical $\Theta_{high}/PV_{low}$ air (average of $[CO]=76 \pm 5$ ppb) and Ex–TP (average of $66 \pm 19$ ppb). This indicates that some of the HBrR air might not have recently entered the stratosphere (within the past few weeks) but some months ($2-3$) prior to the measurements. However, the dominant transport of tropical $\Theta_{high}/PV_{low}$ air to the LMS using CO as a tracer was demonstrated by Hoor et al. (2005), the first of its kind. They concluded that transport of air masses to the LMS over Europe in summer/fall is dominated by transport from the tropical UT/TTL up to $55\%$ above $\Delta\Theta > 25$ K and a slightly smaller contribution from the stratospheric background aged air of up to $45\%$. The influence from the extratropical troposphere is dominant at the TP up to $\Delta\Theta < 25$ K. The study by Bönisch et al. (2009) had similar conclusions.

The persistence of the HBrR observed in many flights during WISE and the already monthly-long presence and respective mixing, generally rules out an origin of this air by some event-like (short duration) recent transport such as by high reaching mesoscale convective systems or other form of STE process. The wide range of lifetimes of these tracers, the distinct change in VMRs across the Ex–TP, the high VMRs of mainly the two longer lived transport tracers extending from the tropical $\Theta_{high}/PV_{low}$ region to the HBrR, as well as the low $O_3$ observed within the HBrR, suggests that the dominant contribution of $Br^{tot}$ to the LMS is via the isentropic transport from the tropical UT/TTL rather than by extratropical STE (e.g., Holton et al.





(1995); Levine et al. (2007); Fueglistaler et al. (2009); Bönisch et al. (2009); Ploeger et al. (2015)). Studies such as by Pan et al. (2016) and Garny and Randel (2016) discuss the transport of Asian monsoon air that is isentropically entrained into the LMS by shedding of small scale eddies. Vogel et al. (2016) provides more evidence that portions of the Asian monsoon anticyclone air mass are transported horizontally to the southwest (via eddy shedding) entering the TTL and further into the LMS in
late boreal summer. A second horizontal transport pathway can occur through filamentation by Rossby wave breaking, east of the anticyclonic event in the extratropics (Vogel et al., 2016). Specific Rossby wave-breaking events producing filaments, irreversibly mix young tropospheric air into older air of the LMS (e.g., Homeyer and Bowman (2013); Müller et al. (2016) and Kunkel et al. (2019)). The relative contributions of these transport pathways are investigated in the next section.

### 4.3    Modelling of the Transport into the LMS

#### 4.3.1    Major Air Mass Origins and Transport Pathways

The tranport of air masses from different atmospheric domains into the NH LMS during summer and fall 2017 is simulated by CLaMS using artificial air mass origin tracers released on 1 May 2017, approximately 5 months prior to the WISE campaign (for details see Sect. 2.5). The advection and mixing processes of the air parcels over the integration time results in the diverse compositions of the LMS air from the various domains of the atmosphere. The fractions of air originating in four of the
atmospheric domains most relevant to our discussion are shown in Fig. 8. In the HBrR (marked by the black box), the fraction of air from the tropical troposphere is $51.2\%$, while only $27.1\%$ is from the stratospheric background and smaller contributions are from the lower TTL ($7.5\%$), mid-latitudinal LMS ($7.0\%$), the mid-latitudinal troposphere ($6.4\%$) and $< 1\%$ from the other domains (see also Fig. 11 (a)). In contrast, the air masses in the LS below and above the HBrR (ranging from equivalent latitudes $55-80°$ N between the TP and $\Theta = 350$ K and $\Theta > 385$ K) consist of $42.6\%$ air from the tropical troposphere, $36.1\%$ from
the stratospheric background, $8.3\%$ from the lower TTL, $7.0\%$ from the mid-latitudinal LMS, $5.2\%$ from the mid-latitudinal troposphere and $< 1\%$ from the other domains (see Fig. 11 (b)).

Over the five months of CLaMS simulation prior to the HALO measurements, the influx of tropospheric air to the HBrR is almost $10\%$ more than the surrounding LS below and above the HBrR. In particular, this increase of tropical tropospheric air ($+8.6\%$) and to a lesser degree extratropical tropospheric air ($+1.2\%$) in the HBrR, results in additional transport of air masses
from these source domains with elevated bromine compared to the surrounding LS (as shown in Fig. 7). Fig. 7 provides a schematic of the relative contributions of tropical $\Theta_{high}/PV_{low}$ air vs the Ex–TP bromine transported to the LMS. The elevated bromine observed in tropical $\Theta_{high}/PV_{low}$ air has air fractions up to $92\%$ (with a mean of $84 \pm 2\%$) originating from the tropical troposphere (Fig. 8 (b)). As expected, air from the tropical troposphere is transported into the tropical $\Theta_{high}/PV_{low}$ region during the last 5 months caused by the large-scale circulation. Therefore, the elevated bromine measured in tropical
$\Theta_{high}/PV_{low}$ air during WISE can be categorized as air masses largely originating in the tropical troposphere as defined by the CLaMS origin tracer. Yan et al. (2019) also showed that similar contributions of air from the $350-360$ K and $370-380$ K layers of $18\%$ and $22\%$, respectively, is transported into the NH LS with only small air mass fractions continuing vertically into the tropical pipe. Their findings are in agreement with those of von Hobe et al. (2021).



Large bromine sources, particularly the VSLS, are from tropical marine regions mainly in coastal areas (Engel and Rigby et al. (2018)). Therefore, surface emission tracers (for details see Sect. 2.5) from CLaMS simulations are also used to infer the contribution of young air masses ($< 5$ months) from different surface regions into the UTLS (see Fig. 9). The surface emission regions with the largest contributions to the HBrR during the WISE campaign flights are shown in Fig. 10. The sum of the surface emission tracers (i.e., air masses younger than $\sim 5$ months) contributing to the HBrR account for $37.7\%$ of the air while the LS region below and above the HBrR both receive less surface air with an average of $30.3\%$ (see Fig. 11 (c–d), respectively, as well as Table 4). Therefore, the larger tropical tropospheric air contribution to the HBrR compared to the surrounding LS as seen in the air mass origin tracers, is largely from recent surface emissions rather than from the free troposphere.

The CLaMS simulations show that surface emissions from the Asian monsoon region (AMR; consisting of NIN, IND, INO, TIB, ECH, and BoB, refer to Fig. 9) and from its tropical adjacent region (TAR; consisting of Wpool, SEA, TWP, TEP, NAF and Neast) have the strongest contributions of the various surface emission tracers to the LMS in late boreal summer and early fall. Maximum contributions of the AMR and TAR surface emissions are delivered to equivalent latitudes between $25-45°$ N and from $\Theta=325-390$ K (see Fig. 10 (a–b)). The grouped emission tracer contributions to the HBrR and surrounding LS can be found in Table 4 and for a breakdown of all surface emission tracers see Fig. 11 (c–d). The AMR accounts for $14.8\%$ of the air in the HBrR, while air originating from the TAR contributes a further $15.2\%$. Together the AMR and TAR make up $30.0\%$ of the air reaching the HBrR, which is $79.8\%$ of all surface air masses. The LS air masses below and above the HBrR receive $\sim 1/4$ less surface (younger) air compared to the HBrR mainly because the AMR and its TAR contribute less.

This is in general agreement with results of CLaMS simulations using a previous version of surface emission tracers for 2012 and 2008 that found a flooding of the northern Ex–LS in boreal summer with young air masses from the region of the Asian monsoon and from the tropical Pacific (Vogel et al., 2016, 2019). Similarly, the Asian monsoon air masses impacting the LMS was observed by Müller et al. (2016). Furthermore, this feature is in general agreement with findings of Orbe et al. (2015), who showed that in boreal summer and fall $\sim 20-30\%$ of the mid-latitudinal LMS air at 100 mb originated from the Asian planetary boundary layer (slightly further north than defined here). The outflow of the AMR into the LMS (see Fig. 10 (a)) has also been recently shown in a study by von Hobe et al. (2021), i.e., the main outflow is below 370 K with weaker outflow up to 380 K. Levine et al. (2007) previously also concluded that vertical transport from the tropical boundary layer into the TTL is strongest over the Indian Ocean, Indonesia and western Pacific Ocean through convection. They further concluded that the main outflow ($\sim 95\%$) of air from the TTL is transported below 380 K into the LMS while only $\sim 5\%$ is transported further upwards into the stratospheric 'overworld'.

A subgroup of regions from the AMR and its TAR belonging to the tropical marine environment (TME; defined by SEA, Wpool, BoB, INO, TWP and TEP) in south-eastern Asia alone contributes $17.7\%$ to the HBrR, which is $47.0\%$ of the air masses from the surface. The maximum TME air mass contribution is in the subtropics to mid-latitudes between $25-55°$ N from $\Theta=320-390$ K, but air masses with elevated TME contributions extend to the tropical UT/TTL and the HBrR (see Fig. 10 (c)). This suggests entrainment of TME air masses crossing the TP around $30°$ N with further transport to the tropical UT/TTL and LMS, in agreement with findings of Orbe et al. (2015). The surrounding LS air masses below and above the HBrR receive





13.8% from the TME, i.e., a 3.9% decrease compared to the HBrR. The variations of TME contribution per flight in the UTLS range from $17.3-30.1\%$ at flight altitude (examples are shown in Figs. 2 (f) and 3 (f) and Fig. 12 (c–d)).

The increased contribution of younger surface air to the HBrR compared to the surrounding air masses below and above, has a significant influence on the trace gas composition. In particular, considering the majority of the surface air masses are transported from regions with large biological activity and known bromine sources (Engel and Rigby et al. (2018)), i.e., the tropical marine environment below the Asian monsoon anticyclone and its tropical adjacent regions. This highlights the influence of surface emissions from south-eastern Asia on pollutants and in particular bromine in the northern LMS during

boreal fall. Adcock et al. (2021) observed enhanced $CH_3Br$ VMRs (up to 9 ppt) as well as $CH_2Br_2$ and H1211 VMRs (up to about 0.75 ppt and 3.5 ppt, respectively; comparable to Engel and Rigby et al. (2018) at the TP in the Asian monsoon region in 2016/2017.

     Using ERA-Interim reanalysis from the years 2007/2008, Vogel et al. (2019) showed that surface air masses can reach 360 K in a few days and reach $360-460$ K within a few months in the tropics and subtropics due to the strong vertical spiral

uplift associated with the Asian monsoon anticyclone including its tropical adjacent air masses. Using data from the ERA-Interim/ERA5 reanalysis, Bucci et al. (2020) showed that air from Nepal in July/August 2017 could rise through convection in the Asian monsoon anticyclone region with transport times up to the TP ($> 17$ km) to be $\sim 20$ days. Similarly, Ashfold et al. (2012) reported a transport time of less than 15 days of air masses from the West Pacific boundary layer ($< 1$ km) up to the lower TTL ($\sim 15$ km). The fast transport (on the order of days) from the Asian tropical planetary boundary layer to the TTL

allows even the short-lived trace substances from the south-east Asian region to reach the UTLS (Orbe et al., 2015).

     The impact of maritime boundary sources to the chemical compostion of the Asian monsoon anticyclone by convective uplift in tropical cyclones in the western Pacific is discussed by e.g. Vogel et al. (2014) and Li et al. (2017, 2021). A study by Liang et al. (2014) concludes that deep convection transports $\sim 7.8-8.4$ ppt brominated VSLS from the marine boundary layer of the tropical Indian Ocean, tropical western Pacific warm pool and the Pacific coast of Mexico to the LMS. These brominated

VSLS are half in the form of source gases and the other half consisting of product gases such as BrO, and only $\sim 10\%$ is lost due to wet scavenging after transport to the TP. Hamer et al. (2020) similarly found that inside the convective system, $CHBr_3$ (one of the VSLS) is the most efficient bromine gas to reach the UT as $> 85\%$ of surface $CHBr_3$ is transported to the tropical UT. In addition, they simulated the presence of 1 ppt of inorganic bromine outside the convective system near Borneo, i.e., in the tropical adjacent regions, while inside the convection system inorganic bromine ($< 0.1$ ppt) is almost completely depleted

due to washout. Therefore, the solubility of the gases and local surrounding conditions may also impact the delivery to the TTL and finally the Ex–LS. This is partially when tropospheric $O_3$ is low and prevents the soluble forms of bromine from reacting and converting to insoluble inorganic bromine compounds.

### 4.3.2   Contributions to UTLS Bromine from Recent Tropical Cyclones

Interestingly, emissions from Central America (CAM) also contributed to elevated $Br^{tot}$ in the tropical $\Theta_{high}/PV_{low}$ air masses

with maximum CAM air mass fractions around $\Theta \sim 345-370$ K during WISE (see Fig. 10 (d)). There are two flights with particularly large air mass fractions from CAM emissions observed during the campaign, i.e., RF08 and RF14 (discussed


below) with flight segments in tropical $\Theta_{\text{high}}$/PV$_{\text{low}}$ air. During both flights, remnant air masses of two mesoscale convective systems in the Central American tropics were probed. For reference for the following discussion, the average contribution of emissions from CAM (excluding RF08 and RF14) per flight ranges from $4.3-8.9\%$ at flight altitude. Generally, the largest

CAM emissions during the WISE campaign are seen in tropical $\Theta_{\text{high}}$/PV$_{\text{low}}$ air with a strong decrease in the LS. The average CAM emissions in the LS range from $2.7-6.2\%$, with the exception of air masses during RF14 and RF15 (also discussed below). Although larger CAM air mass fractions contribute to elevated bromine in tropical $\Theta_{\text{high}}$/PV$_{\text{low}}$ air, the HBrR only has $3.3\%$ of the air fraction from CAM similarly to the LMS which has mostly low CAM air mass contributions.

The first flight with large segments of elevated contributions of air masses from CAM was RF08 on 1 Oct. 2017 (Fig. 3 (f)).

This flight, off the coast of Ireland over the Atlantic Ocean (up to $15.0°$ W), encountered the remnants of Hurricane Maria which had transitioned into an extratropical cyclone and later dissipated on 2 Oct. 2017 (Pasch et al., 2019). The average CAM surface air fraction along the flight track is $18.0 \pm 13.0\%$ with the maximum contribution reaching $60.6\%$ at 20:35 UTC at $\Theta=361$ K and $\Delta\Theta=-11$ K (Fig. 3 (a) and (f)). However, sections with increased CAM air mass fractions only reached the UT while the two flight sections breaching into the LS have low CAM emissions. This low pressure system transported surface air

with increased CAM air mass fractions ($25-45\%$), CO, and N$_2$O to the UT, bringing with it more brominated VSLS up to $[\text{Br}^{\text{VSLS}}]=5.8$ ppt (at 17:29 UTC in Fig. 3 (g)), coinciding with flight sections in tropical $\Theta_{\text{high}}$/PV$_{\text{low}}$ air (light gray shaded sections). These flight segments in tropical $\Theta_{\text{high}}$/PV$_{\text{low}}$ air with large CAM air mass fractions additionally have $\sim 20\%$ of the air mass fraction from the TME, resulting in the UT $[\text{Br}^{\text{org}}] > 19.2$ ppt and $[\text{Br}^{\text{tot}}]=20.7-23.4$ ppt between 15:35–17:41 UTC. The combination of relatively large air mass fractions from CAM and TME in tropical $\Theta_{\text{high}}$/PV$_{\text{low}}$ air, results in some of the

most-bromine-rich air observed during WISE. Even though the tropospheric portions of the flight had elevated CAM emissions, the LS sections only have an average CAM air mass fraction of $3.1 \pm 1.7\%$ and low Br$^{\text{tot}}$ ($17.1-18.9$ ppt below the LS average indicated with the gray dotted line in Fig. 3 (g)). Only a short flight segment reached the HBrR, which is indicated by the dark gray shading in Fig. 3, where the CAM air mass fraction is low ($\sim 1\%$) while the TME air mass fraction is still $\sim 11\%$.

A second tropical cyclone encountered during WISE was Hurricane Ophelia. It made landfall over Ireland on the 16 October

2017 and dissipated over Norway two days later and rapidly transported tropical air from Central America to Europe (Stewart, 2018). This is verified from data collected during RF14 on the 15 October 2017 (Fig. 12 left column). A large influence of surface emission tracers from CAM is present with a flight average of $23.3 \pm 11.8\%$ (see Fig. 12 (a), (c) and (e)). The initial flight section between $\sim$13:20-15:00 UTC just below $\Theta=355$ K, indicates the transport of recently tropical air masses (low equivalent latitudes) with large CAM air mass fractions. Unlike RF08, this low pressure system transported larger CAM air

masses into the LMS. The average influence from CAM air masses in the LMS flight segments is $20.8 \pm 5.4\%$. These are the largest influxes of CAM emissions into the LMS detected during the WISE campaign. Although this flight does not have organic bromine data available, the variability of air masses probed by the aircraft in the LMS is noticeable in Br$_y^{\text{inorg}}$. For example, between 16:04–16:58 UTC the aircraft flew in the LMS (up to $\Delta\Theta=16.7$ K) and relatively low $[\text{Br}_y^{\text{inorg}}]=2.2 \pm 0.4$ ppt was observed with high CAM ($22.3\% \pm 4.1\%$) and TME ($30.4\% \pm 2.5\%$) air fractions. Shortly after, between 17:04–17:46 UTC

when the aircraft cruised in the UT (around $\Delta\Theta \sim -10$ K), elevated Br$_y^{\text{inorg}}$ was observed and reached up to 7.8 ppt. During this event low CO and N$_2$O mixing ratios, low CAM emissions ($6.2 \pm 2.0\%$), only slightly lower TME air mass fractions





$(24.2 \pm 3.6\%)$ and large air mass lag-times (up to 1 yr old) were observed in the UT. This event characterizes an instrusion of stratospheric air into the UT with $Br_y^{inorg} > 4$ ppt (indicated by the light blue shading).

A third flight, RF15 on 19 Oct. 2017 (Fig. 12 right column), around Ireland and the UK likely still captured former air masses from Hurricane Ophelia. The air of this flight is also characterized by a slightly elevated CAM influence in the LS mainly with a surface emission of $8.9 \pm 4.1\%$ at extratropical equivalent latitudes (Fig. 12 (c–d)). Only partial organic bromine data is available with slightly elevated VMRs (Fig. 12 (f)). $CH_3Br$ VMRs up to 7.8 ppt and bromine in halons up to 8.1 ppt are observed in the LMS ($\Delta\Theta=22$ K) with elevated CAM emissions around $14\%$ (and TME air mass fractions $\sim 27\%$). A similar event is again detected during the intial ascent between 09:25–09:38 UTC (light blue shading) after passing through a small stratospheric intrusion. Elevated $Br_y^{inorg} > 4$ ppt and up to 9.2 ppt was observed in the UT ($\Delta\Theta=-33.2 \pm 3.6$ K) while air mass fractions from CAM emissions were low ($7.4 \pm 1.9\%$) while TME emissions were around $25.0 \pm 1.8\%$.

In summary, the synoptic-weather-sytem-related contributions from the CAM region evidently may directly impact observed UTLS bromine. The CLaMS simulations indicate the variety of different air masses influencing our observations. Near the tropical TP, high CAM air mass fractions (through rapid vertical transport by hurricanes), as well as high TME air mass fractions (uplifted by local convection associated with the Asian summer monsoon), result in significantly elevated $Br^{tot}$ observed in the former tropical $\Theta_{high}/PV_{low}$ air (RF08). However, in the LMS there are lower air mass fractions from CAM but still relatively high TME contributions which also result in elevated $Br^{tot}$, i.e., the HBrR during eight flights. Additionally, the two events with higher $Br_y^{inorg}$ observed in the UT (shaded light blue sections in Fig. 12), with comparatively lower CAM emissions and low CO and $N_2O$ mixing ratios, are of particular interest. These events may indicate that air from the LMS intruded into the UT by a recent STE event, which is discussed in a recent study by Kunkel et al. (2019).

## 4.4 Consequences for LMS $O_3$

The isentropic transport of bromine-enriched air from tropical UT/TTL into the LMS likely has a significant impact on $O_3$. In the HBrR, the average $O_3$ measured is $301 \pm 69$ ppb (with a range of $153-524$ ppb), while the LS below and above has an average of $404 \pm 169$ ppb (range $73-791$ ppb) largely due to the significant increase of $O_3$ further into the stratosphere that extends into the upper area of the HBrR. However, in the HBrR, lower $O_3$ VMRs are measured with a slight increase below the HBrR (see Fig. 5 (e)).

In order to investigate the influence of elevated bromine on the Ex–LS $O_3$, we conducted three simulations using the 3–D global TOMCAT model with 22 months integration time (see Sect. 2.6). The Base run was initialized with constant bromine (19.6 ppt from the Ex–TP) VMRs along the entire TP. Run 1 and 2 each had elevated bromine boundary conditions in the tropical UT in addition to the Ex–TP boundary conditions from the Base run (see Table 3). The run 1 and 2 are initialized with an additional $[Br^{tot}]$ of 1.2 ppt and 2.1 ppt, respectively in the tropical UT. The increase in run 1 is mainly due to increased $Br_y^{inorg}$, while the increase of $Br^{tot}$ in run 2 is largely due to additional $Br^{VSLS}$.

These increases of bromine in the tropical UT lead to additional destruction of $O_3$ in the LMS. It can be used as a measure of the $O_3$ sensitivity due to transport of bromine-enriched air from tropical $\Theta_{high}/PV_{low}$ region into the LMS by the lower branch of the Brewer-Dobson circulation. The TOMCAT $O_3$ VMR distribution over the NH from the Base run is shown in Fig. 13 (a),





and is comparable to the FAIRO measured $O_3$ VMRs (Fig. 5 (e)). The relative changes of TOMCAT $O_3$ between the Base run and either run 1 or 2 are displayed in Fig. 13 (b–c), respectively. The impact of increasing tropical UT bromine (run 1), leads to $5.0 \pm 0.4\%$ more $Br^{tot}$ in the HBrR compared to the Base run which results in an $O_3$ change of $-2.5 \pm 0.2\%$ (panel (b)). Run 2 leads to $8.7 \pm 0.7\%$ more $Br^{tot}$ in the HBrR compared to the Base run which results in an $O_3$ change of $-2.8 \pm 0.2\%$ (panel

(c)). In the HBrR, the absolute $O_3$ mixing ratios change by $-6.6 \pm 1.1$ ppb (run 1) and $-7.4 \pm 1.5$ ppb (run 2) (not shown). The decrease of $O_3$ is not strictly linear with increasing $Br^{tot}$, partially due to the nature of the increased bromine ($Br^{VSLS}$ vs $Br_y^{inorg}$) differing between run 1 and 2, as well as other feedback loops such as via lower stratospheric $HO_x$ chemistry.

The average relative $O_3$ loss throughout the whole LS is $-2.6 \pm 0.6\%$ and $-2.7 \pm 0.4\%$ for run 1 and 2, respectively. However, the largest relative $O_3$ decreases are observed along the tropical TP where there is the largest bromine enhancement

due to the intialization. Further significant decreases of $O_3$ extend to the mid-latitudes along the TP. Run 1 has the strongest $O_3$ decreases up to $-5.4\%$ along the TP in the tropics to $40°$ N equivalent latitude (in comparison to the Base Run). As the bromine increase in run 1 is mainly in the form of $Br_y^{inorg}$, this shows the efficiency and immediate destruction of $O_3$ by $Br_y^{inorg}$. Run 2 has a maximum $O_3$ decrease of $-3.9\%$, however the strongest decreases are not near the tropical TP but rather along the mid-latitude TP. As the bromine increase of run 2 is largely in the form of $Br^{VSLS}$, $O_3$ is not immediately destroyed

because of the time delay for the conversion of $Br^{org}$ to $Br_y^{inorg}$. Although the $O_3$ destruction is not as strong near the source of increased bromine in run 2 compared to run 1, the $O_3$ decrease extends further towards the pole just above the TP. The $O_3$ decrease along the TP up to $\Delta\Theta < 20$ K results in $-3.2 \pm 0.6\%$ and $-3.0 \pm 0.4\%$ for run 1 and 2, respectively (available data only between equivalent latitudes of $22-88°$ N).

A study by Hossaini et al. (2015) analyzed the impact of increasing VSLS (bromine, chlorine and iodine) on $O_3$ in the

global LS. The model results showed LS $O_3$ decreases of $4-12\%$, where the brominated VSLS account for $\sim 85\%$ of the $O_3$ destruction. In TTL, the $O_3$ impact just below 100 hPa was $\sim -8\%$ with stronger decreases towards the TP and surface (Hossaini et al. (2015) supplement Fig. S1). At 200 hPa at the North Pole the relative $O_3$ change was $\sim -5\%$, also with stronger decreases at lower altitudes. These are comparable relative $O_3$ percentage changes to our study when considering only the bromine VSLS impact as well as a significantly larger difference of the VSLS bromine loading (up to 8 ppt) in their model

runs. If the transport of source gases into the stratosphere in the tropics increases as climate change progresses, the strongest decrease in $O_3$ can be expected near the cold point TP in agreement with the simulations here (e.g., Hossaini et al. (2012a); Falk et al. (2017) and Ziska et al. (2017)).

## 5 Conclusions

We report on simultaneous measurements of $O_3$, $NO_2$, BrO, $Br^{org}$, $Br_y^{inorg}$, and $Br^{tot}$, as well as transport tracers (CO

and $N_2O$) and a tracer for the mean age of air ($SF_6$), from the HALO aircraft during the WISE campaign of fall 2017 in the NH UTLS. Additionally, CLaMS simulations of $O_3$, $NO_2$, and BrO are reported with generally good agreement to the measurements. The majority of the observations vary between $8.5-14.5$ km, reaching a potential temperature of $\Theta \sim 410$ K,





and focussing on the LMS. The observations include air masses of the entire NH in equivalent latitudes indicating geographical distributions as well as allowing for transport interpretation.

$Br^{tot}$ is assessed from simultaneous measurements of all relevant $Br^{org}$ source gases and inferred $Br_y^{inorg}$. The $Br^{org}$ source gases include $CH_3Br$, the four halons, and the five major brominated very short-lived substances. $Br_y^{inorg}$ is inferred from remotely measured $BrO$ scaled by the modelled $BrO/Br_y^{inorg}$ partitioning. The resulting LS weighted average is $[Br^{tot}]{=}19.2 \pm 1.2$ ppt. Our observations largely agree with previous measurements of $Br^{tot}$ in the UTLS (e.g., Engel and Rigby et al. (2018)), but emphasize some of the causes for the variability in $Br^{tot}$ in the Ex–LS. Air mass lag-times inferred

from $SF_6$ measurements vary by over 3 years.

Major insights into the causes of the $Br^{tot}$ variability have been gained from studying the geographical distribution of the brominated gases as well as transport tracers as a function of equivalent latitude, both from observations (Fig. 5) and transport modelling (Figs. 8 and 10). The key findings of our study are summarized in Fig. 7. Elevated $Br^{tot}$ is observed in a persistent region extending into the LMS, i.e., HBrR* with $[Br^{tot}]{=}20.9 \pm 0.8$ ppt.

The air mass origin tracers modelled by CLaMS agree reasonably well with our observations. They indicate that the HBrR in the LMS has $51.2\%$ of air originating from the tropical troposphere and $6.4\%$ from the mid-latitude troposphere which are mixed with only $27.1\%$ stratospheric background air (as well as contributions from other domains). The LS air masses below and above the HBrR, show a decreasing tropospheric contribution while an approximately equivalent increasing stratospheric background contribution. This increase of approximately $10\%$ young (tropical and extratropical) tropospheric air in the HBrR

compared to the LS below and above it, is reflected in the bromine-rich air masses of the HBrR.

Additional CLaMS surface-emission simulations indicate the majority of surface contributions to the HBrR are from the Asian monsoon region (AMR) and its tropical adjacent region (TAR) which account for $30.0\%$ of the air mass in the HBrR. Only a small fraction of air ($3.3\%$) in the HBrR originates from Central America (CAM). The LS below and above the HBrR only receive $23.2\%$ from the AMR and TAR. This shows the efficiency of the vertical uplift of the AMR rapidly transporting

surface air from the TAR and specifically the tropical marine environment (TME) into tropical UT/TTL and further via the lower branch of the Brewer-Dobson circulation into the Ex–LS. This transport pathway in boreal late summer and fall has a direct impact on the trace gas composition and specifically of bromine in the northern LMS. Our study complements previous studies on the importance of the transport of air masses via the lower branch of the Brewer-Dobson circulation (e.g., Hoor et al. (2005); Levine et al. (2007); Bönisch et al. (2009); Vogel et al. (2016) and Hauck et al. (2020)).

However, during WISE we observed additional causes for varying bromine in the UT. Notably, air masses of high bromine were transported by two hurricanes from CAM into the tropical $\Theta_{high}/PV_{low}$ region, of which the remnants were later observed to the west of the British Isles on 1 October 2017 (RF08). These large CAM (and TME) contributions lead to the bromine-richest air masses observed during WISE with $[Br^{tot}]{=}21.6 \pm 0.7$ ppt in the $\Theta_{high}/PV_{low}$ region. Additionally, our flights captured two STE events where stratospheric air of elevated $Br_y^{inorg}$ was recently transported into the UT caused by breaking

Rossby waves in the extratropics.

Transport of younger bromine-rich tropospheric air masses into the LMS, directly impacts LS $O_3$. The consequences of more $Br^{tot}$ in the LS on $O_3$ is analyzed with TOMCAT model simulations. An enhancement of $\sim 1{-}2$ ppt $Br^{tot}$ in the tropical



UT and further in the LS through isentropic transport, results in the LS $O_3$ change of $-2.6 \pm 0.7\%$ (average of two elevated bromine scenarios) conveying its sensitivity to bromine. The $O_3$ change in the 20 K layer directly above the TP is $-3.1 \pm 0.7\%$.

The continuous nature of our measurements covering a large range of air mass lag-times even for individual flights, offers several advantages over previous mainly balloon and some aircraft-related studies on $Br^{tot}$. This includes an improved space and time resolution as well as a higher precision and accuracy of inferred $Br^{tot}$. Another major advantage of the aircraft measurements is the ability to display a wide-spread behaviour of $Br^{org}$, $Br_y^{inorg}$, and $Br^{tot}$ in the investigated air masses. The multi pathway transport of bromine (as well as of other gases) into the northern LMS via the major branches may well be very

specific for the region and season (mid-latitudes in boreal fall) investigated in the present study. However, it lends evidence that bromine as well as other $O_3$-harming gases are not simply injected into the stratosphere from a single source region, but that a manifold of different pathways may act (e.g., Holton et al. (1995); Homeyer and Bowman (2013); Ploeger et al. (2015); Kunkel et al. (2019)). In this respect, further probing the TP region and LS for $Br^{tot}$ and other $O_3$ depleting species is required. Ultimately, the detected variability of $Br^{tot}$ in the UTLS may not give a firm answer on $Br^{tot}$ and its trend in the middle and

upper stratosphere. Therefore, probing the middle and upper stratosphere for $Br^{tot}$ (and other $O_3$ depleting substances) by ground-based, balloon-based instrumentation, as well as satellites is still required in the future.

*Data availability.* Most data is available on the HALO data depository (https://halo-db.pa.op.dlr.de/mission/96): 7118-7147 (mini-DOAS BrO and $NO_2$ data), 5669-5698 (FAIRO $O_3$ data), 5616-5653 (DLR BAHAMAS aircraft data), 5974-5988 (UMAQS CO and $N_2O$ data), 5556-5585 (GhOST organic bromine and $SF_6$ VMR and mean air mass lag-time data), and 5449-5480 (CLaMS equivalent latitudes and air

mass origin tracers). These data sets are accessible upon signing a data protocol. The surface emission tracer and trace gas curtains from CLaMS as well as the TOMCAT simulations are available upon request.

*Author contributions.* MR, TH, FK, BS and KP operated the mini-DOAS instrument; AE, TK and TS provided the GhOST measurements; AZ and his group provided FAIRO $O_3$ data; PH and VB provided the UMAQS data; JUG and BV provided the CLaMS simulations; and MC provided the TOMCAT model results. MR performed the data analysis and wrote the paper with contributions from KP.

*Competing interests.* The authors declare that they have no conflict of interest.

*Disclaimer. Financial support.* The study was funded by the German Research Foundation (Deutsche Forschungsgemeinschaft; DFG) HALO-SPP 1294. The contributions from MR, TH, and KP were supported via the DFG grants PF 384/16, PF 384/17 und PF 384/19; AE, TK and TS via DFG projects EN 367/11, EN 367/13, EN 367/13-1 und EN 367/14-1; PH and VB via DFG projects HO 4225/7-1 and HO 4225/8-1; and BV via DFG project VO 1276/5-1. Additional financial support by the German Federal Ministry of Education and

Research (BMBF) through ROMIC II SCI-HI project for the partners at the University of Heidelberg (grant number 01LG1908D), University



of Frankfurt (grant number 01LG1908B), and Forschungszentrum Jülich (grant number 01LG1908C) is highly acknoweleged. The modelling work at Leeds was support by the NERC SISLAC project (NE/R001782/1).

*Acknowledgements.* We thank the Deutsches Zentrum für Luft- und Raumfahrt (DLR) for the support to get the instrument certified and the DLR Flugexperimente Team at Oberpfaffenhofen, in particular, Frank Probst, Martina Hierle, Andreas Minikin and Andrea Hausold, for the
support given during the WISE mission. Special thanks to the scientific coordinators of the WISE campaign, Peter Hoor (University Mainz) and Martin Riese (FZJ). For more information about the WISE campaign, visit https://www.wise2017.de/ (last access: 3 June 2020).

The authors gratefully acknowledge the computing time for the CLaMS curtain simulations granted on the supercomputer JURECA at Jülich Supercomputing Centre (JSC) under the VSR project ID JICG11. The TOMCAT/SLIMCAT simulations were performed on the Leeds ARC and national Archer HPC machines.

The authors acknowledge EARLINET for providing aerosol LIDAR profiles available at https://data.earlinet.org/. The research leading to these results has received funding from the European Union's Horizon 2020 research and innovation programme under grant agreement No 654109 and previously from the European Union Seventh Framework Programme (FP7/2007-2013) under grant agreement n° 262254. Additionally, the authors acknowledge the use of SAGE III - ISS aerosol data, which were obtained from the NASA Langley Research Center Atmospheric Science Data Center.





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





**Table 1.** DOAS spectral analysis details for the various trace gases and their spectral fitting intervals. Refer to Table 2 for the corresponding fitted absorbers. The $\sigma$ [dSCD] are typical errors in molecules cm$^{-2}$.

| Gas | Spectral interval [nm] | Fitted absorbers | Add. Param. | Reference | $\sigma$ [dSCD] |
|---|---|---|---|---|---|
| O$_3$–UV | 338–357 | 1, 4, 5, 6, 7 | $^{(i)}I_{Ofs}$, $^{(ii)}$R, $^{(iii)}$R·$\lambda^4$ | Pukīte and Wagner (2016) | 2.5 x 10$^{17}$ |
| BrO | 346–359 | 1, 4, 5, 6, 7 | $I_{Ofs}$, R, R·$\lambda^4$ | Aliwell et al. (2002) | 2.4 x 10$^{13}$ |
| O$_3$–vis | 450–500 | 2, 3, 4, 6, 8 | $I_{Ofs}$, R, R·$\lambda^4$ | Hüneke et al. (2017) | 3.2 x 10$^{17}$ |
| NO$_2$ | 432–490 | 2, 3, 4, 6, 8 | $I_{Ofs}$, R, R·$\lambda^4$ | Hüneke et al. (2017) | 2.8 x 10$^{14}$ |

(i) $I_{Ofs}$: Offset spectrum; (ii) R: Ring spectrum; (iii) R·$\lambda^4$: Ring spectrum multiplied with $\lambda^4$





**Table 2.** Spectral retrieval absorption cross sections used for the DOAS method analysis (Hüneke et al., 2017; Stutz et al., 2017). The solar-$I_0$ correction used is suggested in Aliwell et al. (2002).

| No. | Absorber | Temperature | Solar$-I_0$ | Reference | Uncertainty |
|---|---|---|---|---|---|
| 1 | $O_4$ | 293 K | | Thalman and Volkamer (2013) | 3–4 % |
| 2 | $O_4$ | 273 K | | Thalman and Volkamer (2013) | 3–4 % |
| 3 | $O_3$ | 203 K | $1.00 \times 10^{20}$ | Serdyuchenko et al. (2014) | 2–3 % |
| 4 | $O_3$ | 223 K | $1.00 \times 10^{20}$ | Serdyuchenko et al. (2014) | 2–3 % |
| 5 | $O_3$ | 293 K | $1.00 \times 10^{20}$ | Serdyuchenko et al. (2014) | 2–3 % |
| 6 | $NO_2$ | 223 K | $1.00 \times 10^{17}$ | Bogumil et al. (2003) | 3.4 % |
| 7 | BrO | 223 K | | Fleischmann et al. (2004) | 8–10 % |
| 8 | $H_2O$ | 273 K | | Rothman et al. (2009) | 8 % |





**Table 3.** Bromine boundary conditions near the TP from various research studies for the Ex–TP and the tropical UT (Trop–UT) used to initialize the TOMCAT/SLIMCAT model. The Base run uses the Ex–TP bromine boundary conditions along the entire TP for all latitudes. There are two further runs with elevated TP bromine initializations in the tropics ($30°$ S–$30°$ N, run 1 and run 2) while the extratropics are still set to the Ex–TP bromine boundary conditions as in the Base run.

| Run | Region | $CH_3Br$ | Halons | $Br^{VSLS}$ | $Br_y^{inorg}$ | $Br^{tot}$ |
|------|---------|-----------|---------|--------------|-----------------|-------------|
| **Base** | **Ex–TP** | $7.25 \pm 0.16$ ppt [1a] | $7.80$ ppt [2] | $3.06 \pm 0.56$ ppt [1b] | $1.44 \pm 0.53$ ppt [1b] | $19.6 \pm 0.8$ ppt |
| **Run 1** | **Trop–UT** | $7.25 \pm 0.16$ ppt [1a] | $7.80$ ppt [2] | $3.12 \pm 0.47$ ppt [3] | $2.63 \pm 1.04$ ppt [4] | $20.8 \pm 1.2$ ppt |
| **Run 2** | | $7.25 \pm 0.16$ ppt [1a] | $7.80$ ppt [2] | $5.00 \pm 0.54$ ppt [1c] | $1.69 \pm 0.54$ ppt [1c] | $21.7 \pm 0.8$ ppt |

[1a] Current study all data at the TP: $\Delta\Theta=-10$ K to WMO TP

[1b] Current study Ex–TP: eq. latitude from $40-60°$ N, $\Delta\Theta=-10$ K to WMO TP

[1c] Current study Trop–UT (i.e., $\Theta_{high}/PV_{low}$): $\Theta > 355$ K, PV $< 2$

[2] from WMO (2018) and cited by Keber et al. (2020)

[3] Navarro et al. (2015) average from East and West Pacific

[4] Werner et al. (2017) and Koenig et al. (2017)





**Table 4.** The air mass contributions of the CLaMS surface emission tracers (defined in Fig. 9) in the HBrR and the LS below and above the HBrR (shown in Fig. 11 (c–d) respectively). The HBrR is between equivalent latitudes $55-80°$ N and $\Theta=350-385$ K, while the LS below and above is between equivalent latitudes $55-80°$ N from the TP to $\Theta=350$ K and $\Theta > 385$ K.

| Emission Tracer ($\Omega_i$) | HBrR | LS Below & Above |
|---|---|---|
| All Surface Regions ($\Omega=\sum_{i=1}^{n=24}\Omega_i$) | 37.65 % | 30.28 % |
| Asian Monsoon Region (AMR):<br>    NIN+IND+INO+TIB+ECH+BoB | 14.81 % | 11.37 % |
| Tropical Adjacent Region (TAR):<br>    Wpool+SEA+TWP+TEP+NAF+Neast | 15.23 % | 11.93 % |
| Tropical Marine Environment (TME):<br>    SEA+Wpool+BoB+INO+TWP+TEP | 17.69 % | 13.81 % |



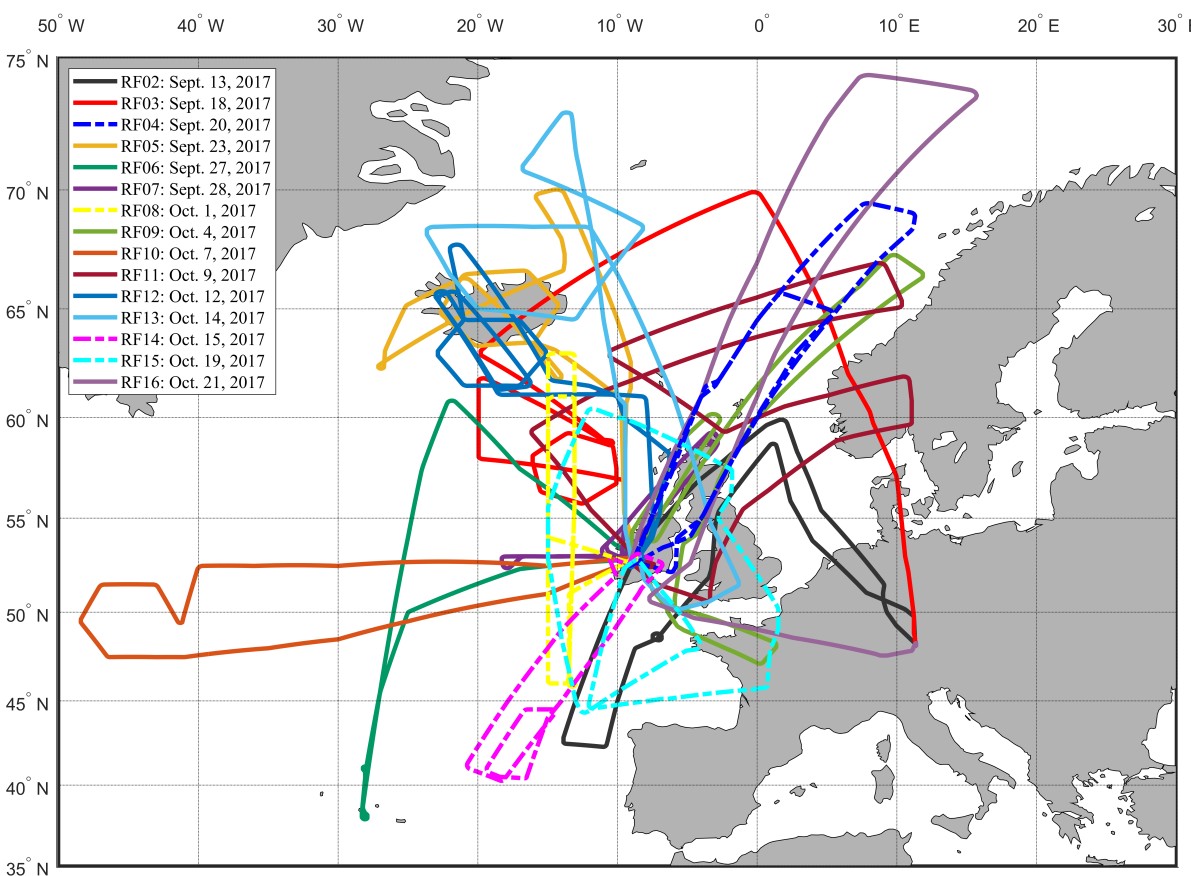

**Figure 1.** Trajectories of the 15 HALO flights during the WISE campaign in the mid-high latitudes over the North Atlantic Ocean, Norwegian Sea and north-western Europe in the UTLS from 13 September 2017 until 21 October 2017. A prior test flight (RF01) is not included in this study. The first research flight, RF02, was conducted locally from the home base in Oberpfaffenhofen, Germany, and all subsequent flights were based from Shannon, Ireland aside from the transfer flights. The flights illustrated by the dashed lines, i.e., RF04 (blue), RF08 (yellow), RF14 (magenta) and RF15 (cyan) are discussed in the text.

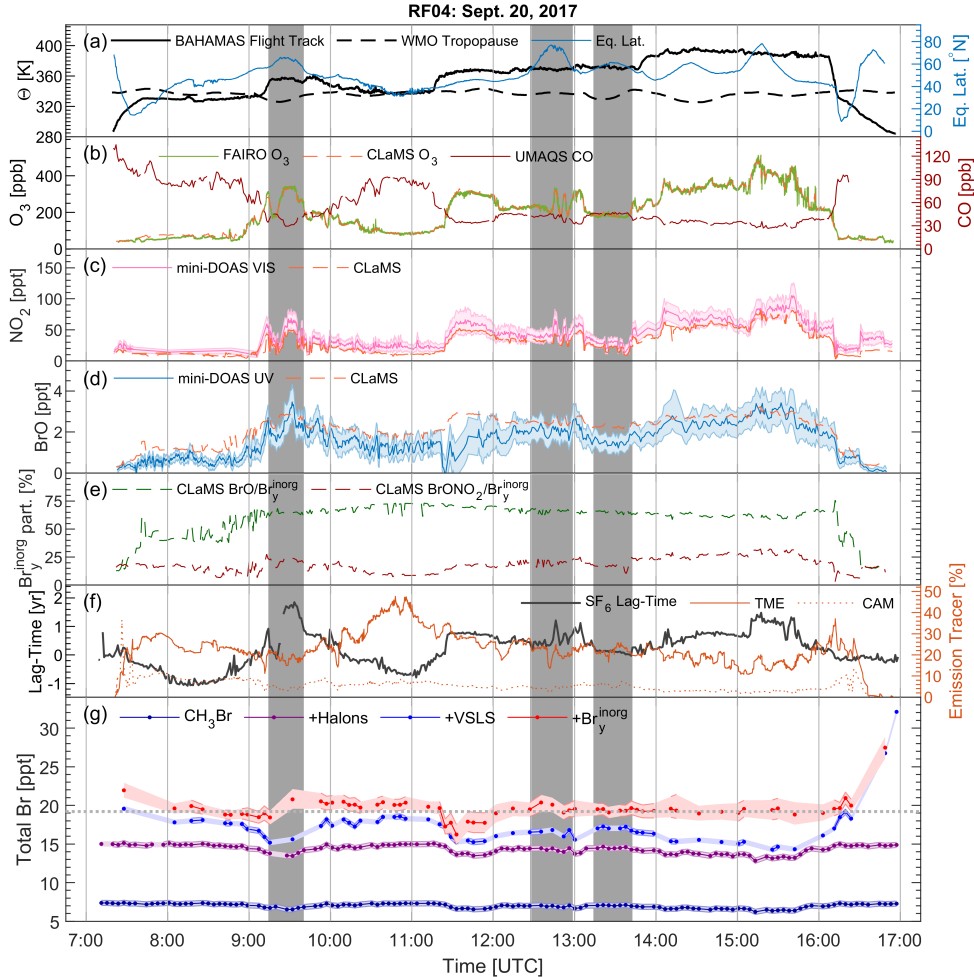

**Figure 2.** Overview of HALO RF04 on 20 September 2017 (blue dashed flight track in Fig. 1). The potential temperature of the flight trajectory and WMO TP (solid and dashed black lines, respectively) and equivalent latitude (solid blue line, right axis) are shown in panel (a). The following panels show (b) FAIRO measured (solid green) and vertically adjusted CLaMS modelled (dashed orange) $O_3$ VMR as well as the UMAQS measured CO (dark red line, right axis); (c) mini-DOAS measured (solid pink) and vertically adjusted CLaMS modelled (dashed orange) $NO_2$ VMR; (d) mini-DOAS measured (solid blue) and vertically adjusted CLaMS modelled (dashed orange) BrO VMR; (e) vertically adjusted CLaMS simulated ratios of $BrO/Br_y^{inorg}$ (dashed green) and $BrONO_2/Br_y^{inorg}$ (dashed dark red); (f) the mean lag-time of air inferred from GhOST's $SF_6$ measurements (black solid line) as well as the CLaMS emission tracers TME (tropical marine environment) from south-east Asia and CAM (Central America) tracer (orange solid and dotted line, respectively on the right axis); and (g) GhOST measured $CH_3Br$ (blue), to which the VMRs of the halons (purple), brominated VSLS (light blue) and inferred $Br_y^{inorg}$ (red) is subsequently added. The gray dotted line in (g) represents the campaign LS weighted average of $[Br^{tot}]=19.2\pm1.2$ ppt. Dark gray shading indicates flight sections in the high bromine region (HBrR) between eq. latitudes $55-80°$ N and $\Theta=350-385$ K, as discussed in the text.

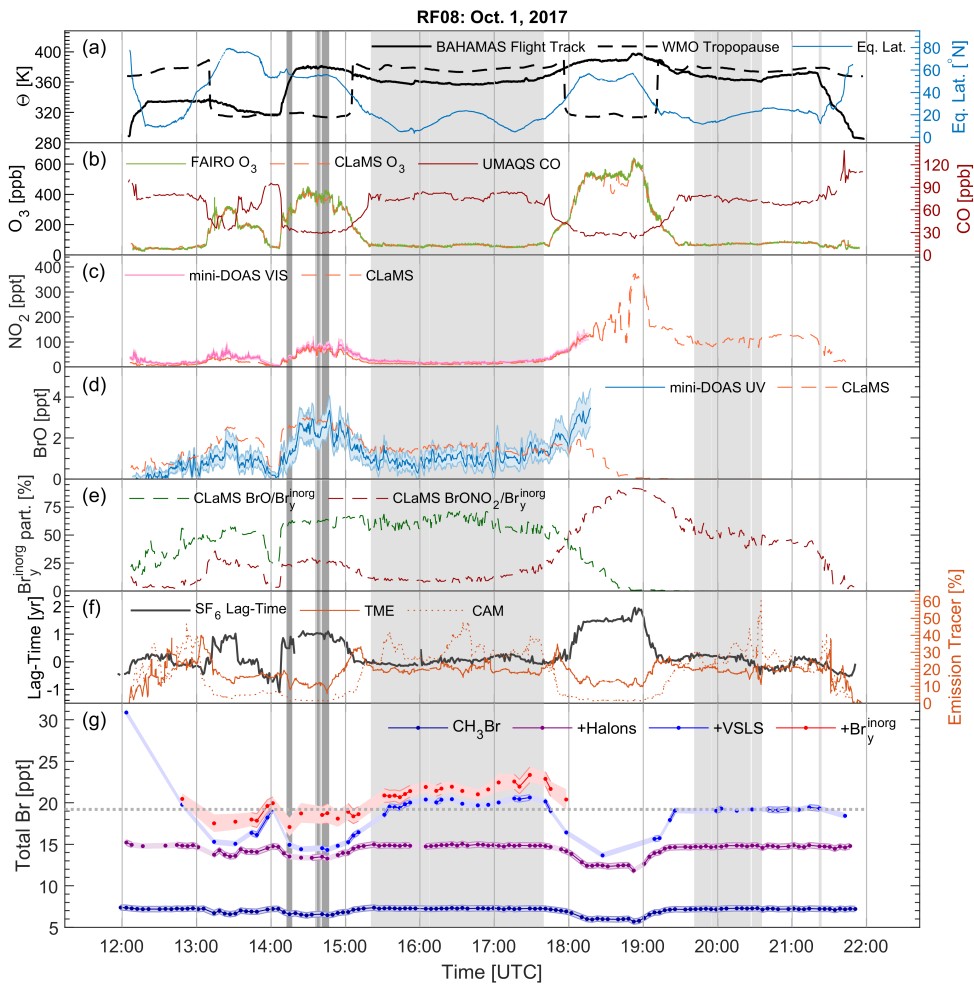

**Figure 3.** Overview of HALO RF08 on 1 October 2017 (yellow dashed flight track in Fig. 1). Panels (a–g) are as given in Fig. 2. Light gray shading indicates flight sections in the tropical $\Theta_{high}/PV_{low}$ region ($\Theta > 355$ K and PV $< 2$, i.e., former tropical UT/TTL air) and the dark gray shading indicates flight sections in the HBrR as discussed in the text. At SZA$> 89°$ mini-DOAS measurements ($NO_2$, BrO and $Br_y^{inorg}$) are unavailable due to the lack in skylight.

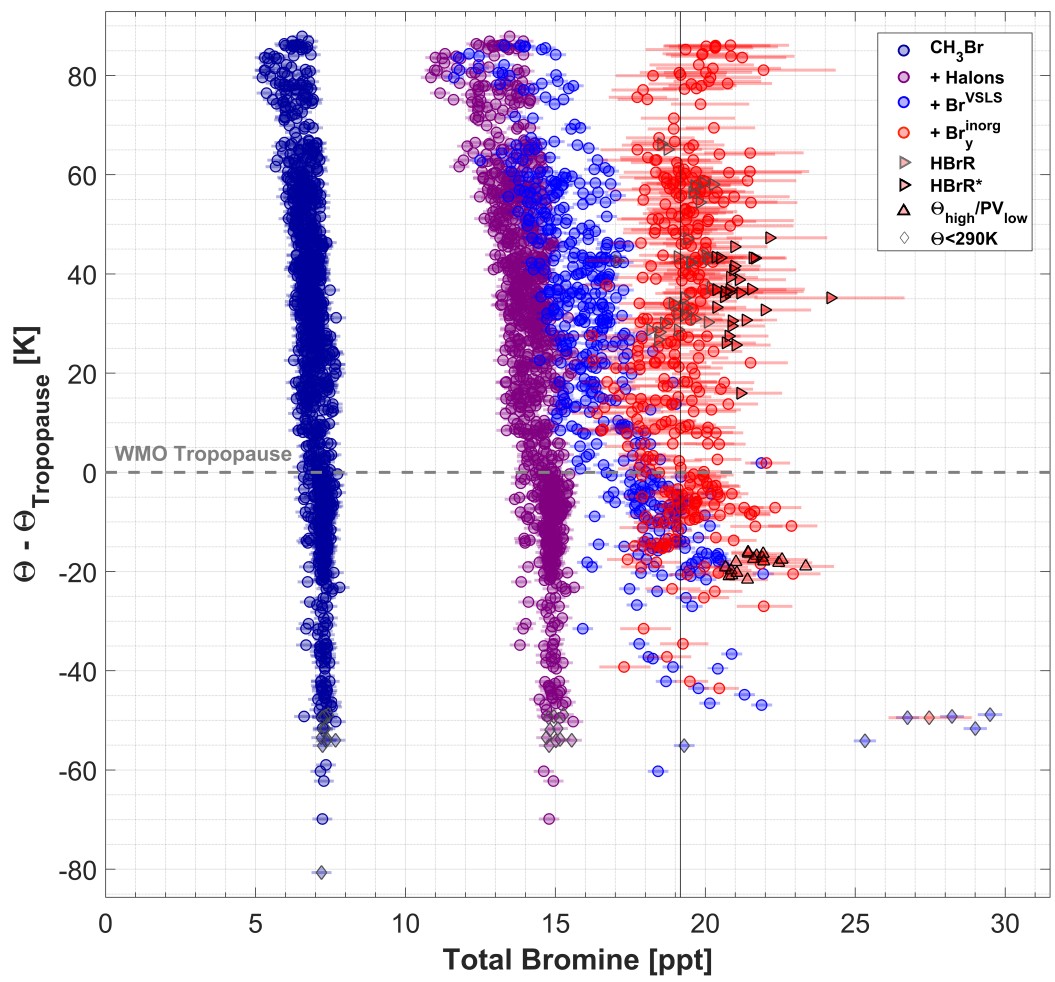

**Figure 4.** Inferred total bromine, $Br^{tot}$, as a function of potential temperature distance from the WMO TP $\Delta\Theta = \Theta - \Theta_{Tropopause}$ [K] during the WISE campaign in fall 2017. The $Br^{org}$ species are summed up according to their Br atomicity: $CH_3Br$ (dark blue), four halons (purple) and brominated VSLS (light blue). The inferred $Br_y^{inorg}$ is subsequently added (red) resulting in the UTLS $Br^{tot}$. The solid black line represents the LS weighted mean $[Br^{tot}] = 19.2 \pm 1.2$ ppt ($1\sigma$ variability). The $Br^{tot}$ measurements from the high bromine region (HBrR) are indicated by right-pointing (gray and black-edged) triangles (weighted mean $[Br^{tot}] = 19.9 \pm 1.2$ ppt). The measurements within the HBrR that are larger than the LS mean by at least $1\sigma$ labeled HBrR*, are indicated by the black-edged right-pointing triangles alone (mean $[Br^{tot}] = 20.9 \pm 0.8$ ppt). Measurements in tropical $\Theta_{high}/PV_{low}$ air are depicted by the upward-pointing black-edged triangles (mean $[Br^{tot}] = 21.6 \pm 0.7$ ppt). Measurements from near the surface during takeoff or landing, i.e., $\Theta < 290$ K, are depicted by the gray-edged diamond symbols which include the VSLS mixing ratios $> 25$ ppt in the lower right corner.

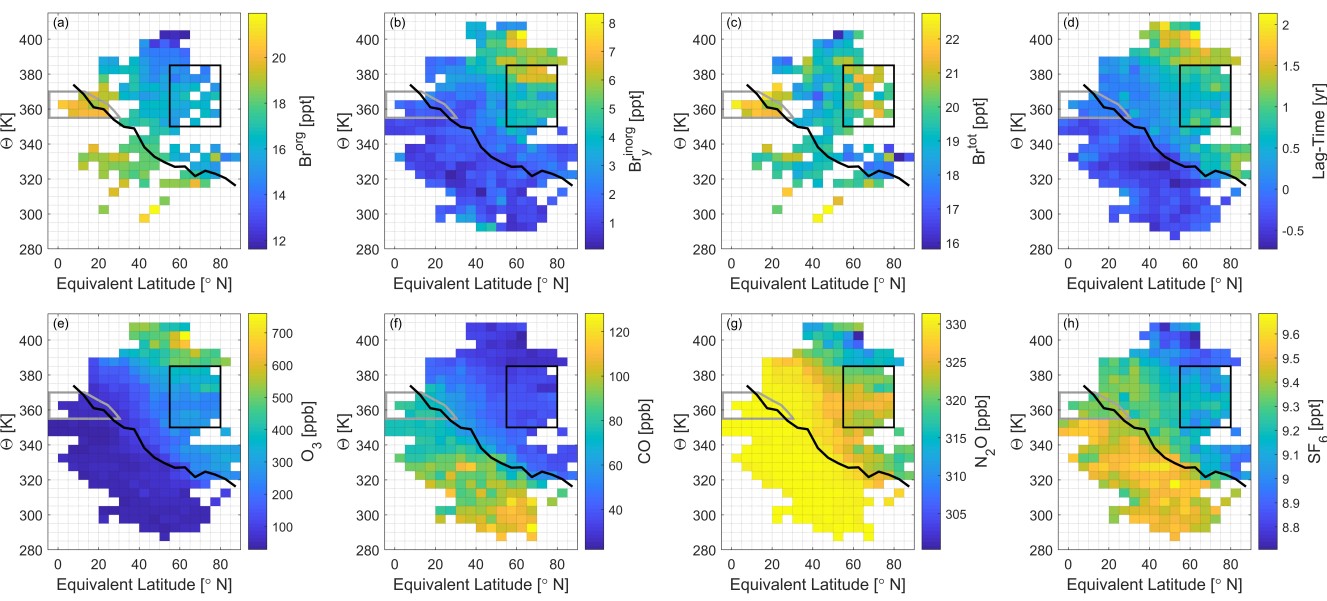

**Figure 5.** The WISE campaign in fall 2017 as a function of equivalent latitude and potential temperature for (a) $Br^{org}$ mixing ratios ($CH_3Br$+halons+VSLS) from GhOST-MS, (b) $Br_y^{inorg}$ VMR from mini-DOAS BrO scaled by the CLaMS $BrO/Br_y^{inorg}$ ratio, (c) $Br^{tot}$ VMR, (d) mean air mass lag-time (based on $SF_6$ data from GhOST-ECD), (e) $O_3$ VMR from FAIRO, (f) CO VMR from UMAQS, (g) $N_2O$ VMR from UMAQS, and (h) $SF_6$ VMR from GhOST-ECD. The data is binned in $5°$ grids of equivalent latitude and 5 K grids potential temperatures. The black solid thick line is the mean WMO TP over the campaign duration. The black box in the LMS from $55-80°$ N and $350-385$ K is the high bromine region (HBrR), and the gray outline at $\Theta > 355$ K and $PV < 2$ marks the tropical $\Theta_{high}/PV_{low}$ region (discussed in the text).





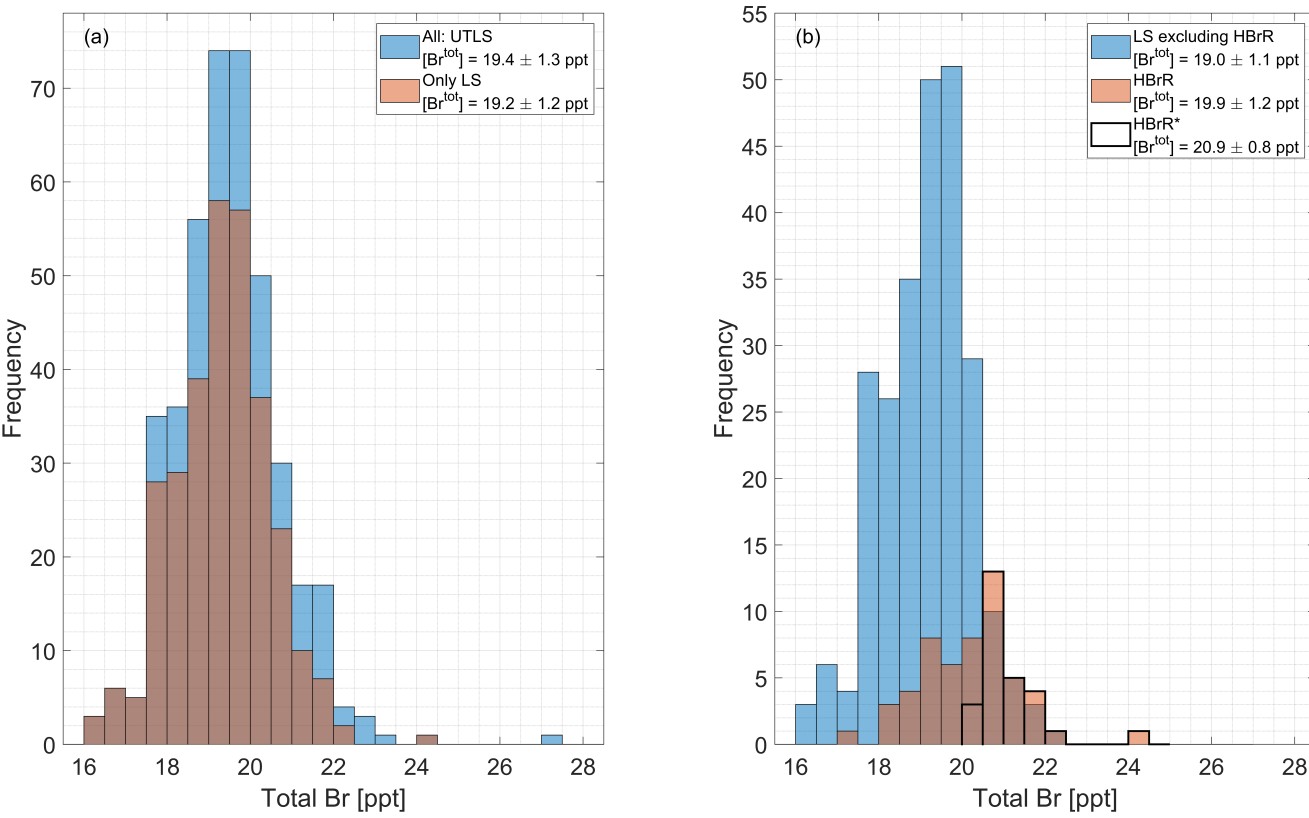

**Figure 6.** Frequency distribution of inferred total $Br^{tot}$. (a) All data (UTLS) have a weighted mean $[Br^{tot}] = 19.4 \pm 1.3$ ppt (blue) and $Br^{tot}$ measurements from the LS have a weighted mean $[Br^{tot}] = 19.2 \pm 1.2$ ppt (brown). (b) Measurements from the LS excluding the HBrR have a weighted mean $[Br^{tot}] = 19.0 \pm 1.1$ ppt (blue) while the HBrR (shown in Fig. 5 (c) by the black box) has $[Br^{tot}] = 19.9 \pm 1.2$ ppt (brown). HBrR* (thick black outlined bars) is the distribution of the measurements in the HBrR that are larger than the LS weighted mean by at least $1\sigma$. HBrR* has a weighted mean of $[Br^{tot}]=20.9 \pm 0.8$ ppt.



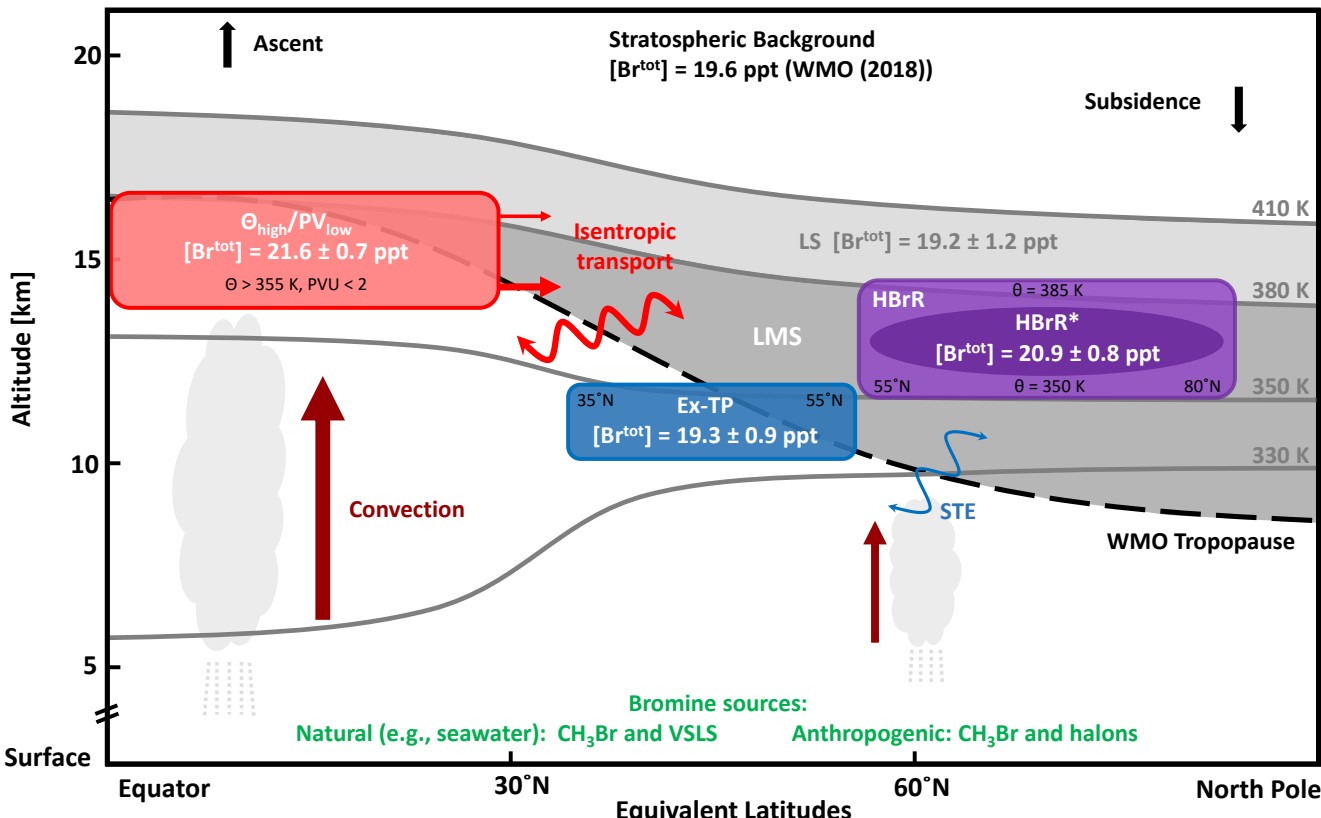

**Figure 7.** Schematic of bromine transport pathways in the UTLS during the WISE campaign in fall 2017. The black dashed line is the WMO TP. The gray lines are constant potential temperature isentropes. The 380 K isentrope marks the upper boundary of the LMS (shaded dark gray region), the 410 K (our measurement upper limit) isentrope marks the upper boundary of the LS (shaded dark + light gray regions). Bromine-rich former tropical UT/TTL air masses at $\Theta > 355$ K and PVU $< 2$ are labelled $\Theta_{high}$/PV$_{low}$ (shaded red box). The Ex–TP (shaded blue box) is defined here as $\pm 5$ K from the local TP between $35-55°$ N. The transport of bromine from these two regions contributes to the high bromine region (HBrR; shaded light purple box) in the LMS, located between $55-80°$ N and $\Theta = 350-385$ K. The Br$^{tot}$ inferred measurements larger than the LS mean by more than $1\sigma$ within the HBrR are indicated by HBrR* (shaded dark purple oval). The dark red arrows show regions of strong convection transporting surface air masses (from bromine source regions) to the UT. The two main transport pathways of bromine are from the tropical UT/TTL by isentropic transport (light red arrows) and less by STE across the TP in the extratropics (blue arrow). Br$^{tot}$ VMRs are from the WISE campaign as discussed in the text.

## CLaMS Air Mass Origin Tracers

**Figure 8.** CLaMS air mass origin tracers from WISE flights are initialized 5 months prior to the campaign. The atmospheric model is divided into nine domains, shown are (a) the stratospheric background, (b) the tropical troposphere, (c) the lower TTL ($\Theta=355-380$ K), and (d) the mid-latitudinal troposphere. The remaining domains include the upper TTL ($\Theta=380-425$ K), the tropical pipe, the mid-latitudinal LMS, the polar troposphere, and the polar LMS (not shown). The solid black line is the campaign average WMO TP, the black box indicates the area of the HBrR and the gray outline marks the tropical $\Theta_{high}/PV_{low}$ region (discussed in the text).



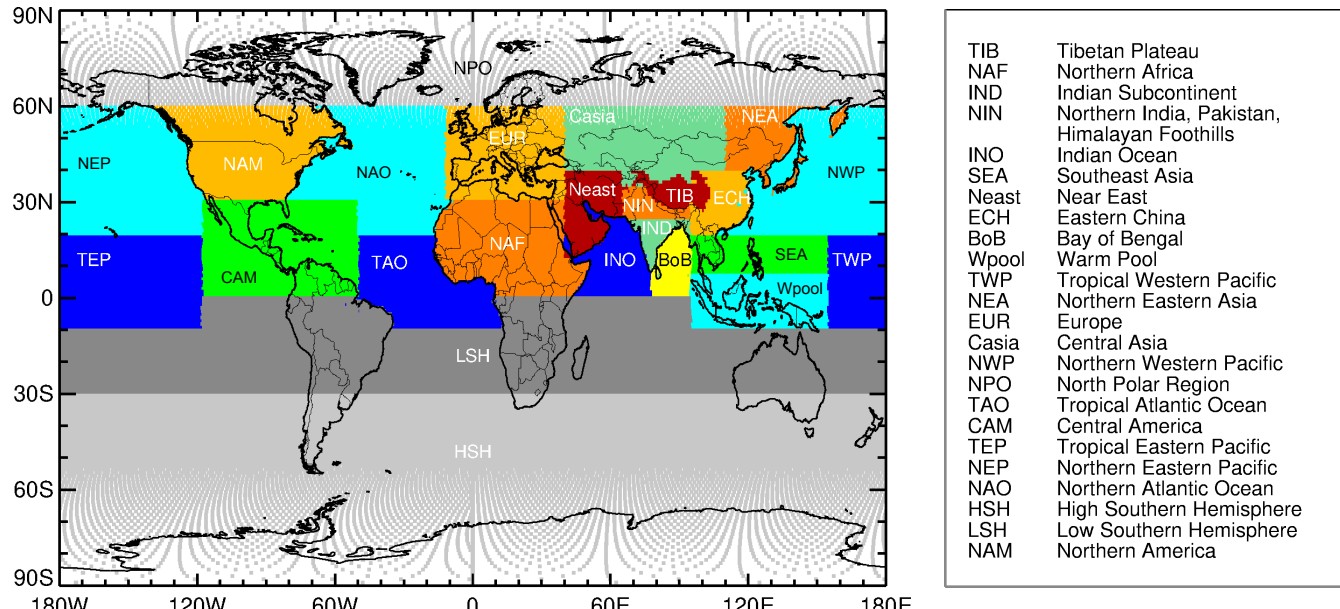

**Figure 9.** CLaMS artificial surface emission tracers by region based on previous simulations described in Vogel et al. (2015, 2019), with higher resolution regions at coastal boundaries mainly in south-eastern Asia (INO, IND, NIN, BoB, Neast, and ECH) as well as in other regions (TIB, NEA, NAF, TAO, NAM, NEP, and NAO). Adapted from Wetzel et al. (2020) where these defined regions were also used for the analysis during the WISE campaign.





**Figure 10.** Relative contribution of surface emission tracers during the WISE campaign. The individual surface regions are displayed in Fig. 9. The Asian monsoon anticyclone region (AMR) consisting of NIN, IND, INO, TIB, ECH, and BoB (panel (a)), as well as its tropical adjacent region (TAR) consisting of Wpool, SEA, TWP, TEP, NAF, and Neast (panel (b)), contribute the largest fractions of surface air to the LMS HBrR (indicated by the black box, see text for details). A subgroup of these regions, the tropical marine environment (TME consisting of SEA, Wpool, BoB, INO, TWP, and TEP) is particularly significant due to elevated emissions of brominated VSLS (panel (c)). The contribution from Central America (CAM) to the tropical UT/TTL is largely due to vertical transport of air by hurricanes Maria and Ophelia in fall 2017 (panel (d)). The gray outline marks the tropical $\Theta_{\text{high}}/PV_{\text{low}}$ region and the solid black line is the average WMO TP.



**Figure 11.** Relative contribution of the air mass origin (a–b) and surface emission (c–d) tracers during the WISE campaign obtained from CLaMS simulations. Contributions to the HBrR (between $\Theta = 350 - 385$ K and equivalent latitudes $55 - 80°$ N) are shown in panels (a) and (c), while the contribution to the LS air below and above the HBrR (i.e., from the local TP to $\Theta = 350$ K as well as $\Theta > 385$ K and between equivalent latitudes $55 - 80°$ N) are displayed in panels (b) and (d).

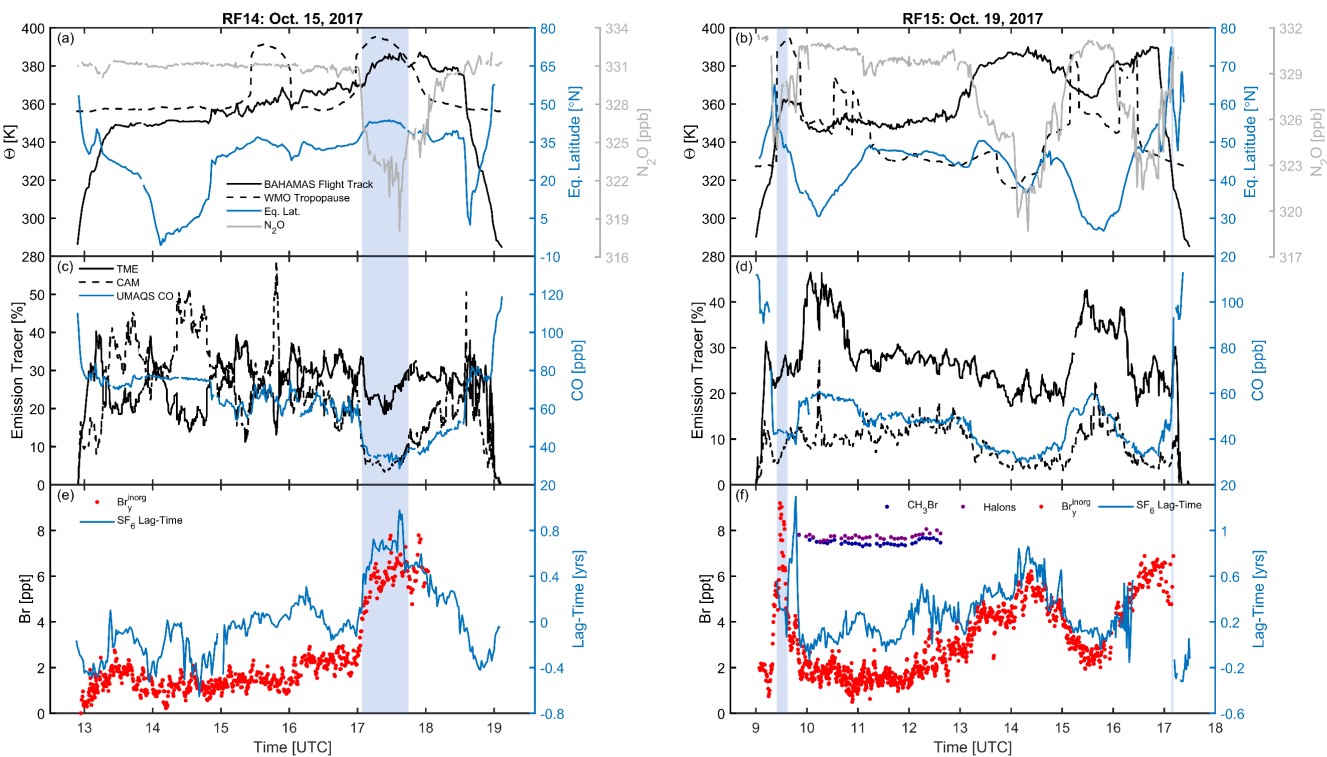

**Figure 12.** Examples of recent air mass surface emissions from the tropical marine environment (TME) in south-eastern Asia as well as elevated emissions from Central America (CAM) during flights in remnants of tropical cyclones. The left column shows details of RF14 on 15 Oct. 2017 and the right column is of RF15 on 19 Oct. 2017. Panels (a–b) show the potential temperature of the flight track (black solid line) and WMO TP (black dashed line) for each flight along with the equivalent latitude (blue line, right axes) and $N_2O$ VMR (gray, right axes). Panels (c–d) show the air mass fraction at flight altitude from the TME and CAM emission tracers (black solid and dashed lines, respectively), as well as the CO VMR (blue, right axes). Panels (e–f) show the individual bromine mixing ratios of $CH_3Br$ (dark blue points), the halons (purple points), and $Br_y^{inorg}$ (red points) where available along with the air mass lag-time (blue line, right axes). The light blue shading indicates $Br_y^{inorg} > 4$ ppt in the UT, i.e., $\Delta\Theta < 0$ K.



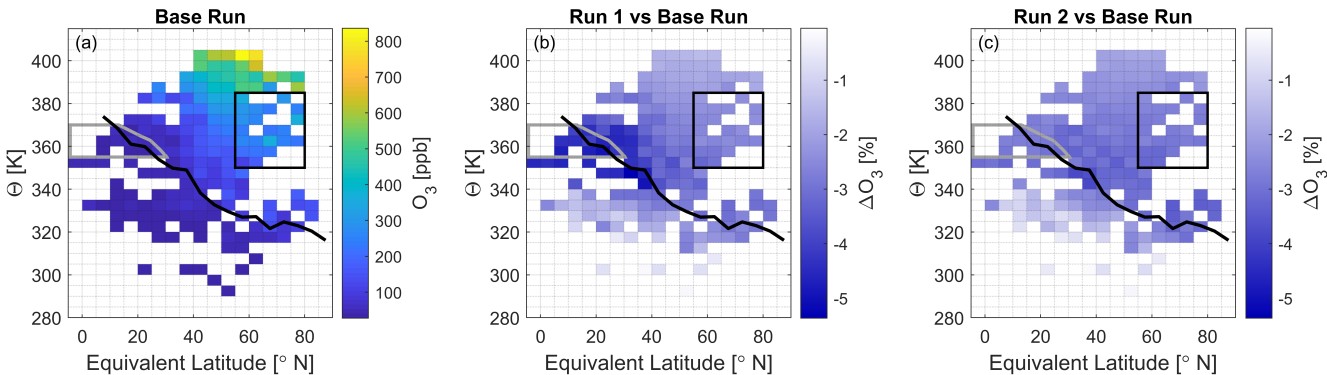

**Figure 13.** TOMCAT model results of $O_3$ impacted by elevated $Br^{tot}$ in the UTLS. Panel (a) displays the $O_3$ VMR distribution from the Base run using constant $Br^{tot}$ boundary conditions along the TP (see Table 3). Panel (b) shows the relative precentage $\Delta O_3$ between run 1 with elevated $Br^{tot}$ in the tropical UT and the Base run. Panel (c) displays the relative $\Delta O_3$ between run 2 and the Base run. The solid black line is the campaign average WMO TP, the black box in the LMS is the HBrR and the gray outline marks the tropical $\Theta_{high}/PV_{low}$ region.