# Peer review of "Organic and inorganic bromine measurements around the extratropical tropopause and lowermost stratosphere: Insights into the transport pathways and total bromine"

_Atmospheric Chemistry and Physics, 2021_

## Author Response (AR1)

Reviewer 1: June 11, 2021

We are grateful for the reviewer's constructive comments and their positive feedback and recommendation for publication. We have included their input for further improvements of our manuscript in the following way.

The reviewer's comments are written in **bold**, our responses are marked with AC (authors comments).
Comments:
**Page 5, lines 146-147: Why only the limb viewing mode of the mini-DOAS instrument is used in this study? The nadir mode could also provide valuable information about the BrO column density above the varying altitude of the aircraft.**
AC: The scaling method is only applicable in the limb viewing geometry. Therefore, the Nadir mode could only provide slant column densities (instead of mixing ratios) where the vertical column amounts only represent the average below the aircraft. The ascent and descent profiles of the aircraft could provide altitude dependent vertical column density information from Nadir observations but they form only a minor part of an individual deployment of the aircraft measurements. Since the present study is mostly devoted to UTLS bromine, within this study only the limb (at flight altitude) BrO mixing ratios are evaluated.

**Page 6, lines 179-181: How is initialised the McArtim RT model for the calculation of NO2 and O3-vis reference SCDs? In a similar way as for the radiative transfer simulations of the limb measurement (see lines 186-190)? This information should be added.**
AC: The McArtim model is initialized the exact same way as the limb α factor calculations, i.e. from simulated CLaMS curtains of the respective gases. We have included the following sentence at the end of the paragraph in line 182: "The initialization of the McArtim model is the same as for the scaling α factor simulations, i.e., from simulated CLaMS curtains of the respective gases."

**Page 7, lines 201-202: Does the scaling method fully compensate for any cloud effect? Did you make any selection of the mini-DOAS limb measurements regarding the cloud conditions by removing the more problematic cloudy scenes in terms of RT modelling?**
AC: The calculated α factors of each gas are dependent on the aerosol and cloud conditions, however the ratio of the α factors of the target and scaling gas have negligible impacts by clouds. This, at first sight astonishing conclusion, is based on the equivalence theorem in optics (Irvine, 1964, reference is provided in the manuscript). In fact, a previous study (Knecht, 2015) compared the radiative transfer in various aerosol loaded and all sky conditions and concluded that the α factor ratios are largely independent on the optical properties of the atmosphere and mainly dependent on the actual concentration profiles of the target and scaling gas. There is a slight solar zenith angle dependence, but only for measurements at low altitudes below ~5km. Therefore, we do not separate cloudy sky measurements from the clear sky sections (for details see also Stutz et al., 2017; Werner et al., 2017; Hüneke et al., 2017; Kluge et al., 2020, references are given in the manuscript).

**Page 9, lines 269-270: You should provide the list of Aura-MLS and ACE-FTS trace gases/tracers which are used to initialize the CLaMS simulations or at least a reference where this list can be found.**
AC: We included the following details in lines 270-273 to clarify the initialization process of the CLaMS simulations: "The simulations are initialized with Aura-MLS (Microwave Limb Sounder) measurements of $O_3$, CO, HCl and $N_2O$ along with ACE-FTS (Atmospheric Chemistry Experiment - Fourier Transform Spectrometer) satellite data tracer-tracer-correlations of $N_2O$ and $CH_4$, CFC-11, CFC-12, and $NO_y$. $NO_y$

is the sum of the gas phase NO, $NO_2$, $2xN_2O_5$, $HNO_3$, $ClONO_2$, and $HNO_4$. Further details of the initialization procedure are described in Grooß et al. (2014).”

**Pages 13-14, lines 400-427: the discussion is a bit difficult to follow here. Maybe the different bromine contributions (CH3Br, halons, VSLS, inorganic) corresponding to the selected typical delta_theta values (+/-5K, 78-88K) could be included in a table?**

AC: As suggested by the referee, a table is included for clarity and the corresponding lines 419-435 are altered to the following text:

“Within the upper LS region ($\Delta\Theta$=78–88 K) referred to as $LS^{top}$, $Br^{org}$ quickly decreases to 13.5 ± 1.2 ppt as $Br^{org}$ is being converted to $Br_y^{inorg}$ species. The breakdown of the following bromine mixing ratios for $Br^{org}$, $CH_3Br$, halons, $Br^{VSLS}$, and $Br_y^{inorg}$ at the TP and higher up in the $LS^{top}$ layer is indicated in Table 4. $CH_3Br$ (contribution (1) to $Br^{tot}$) is destroyed with increasing distance into the stratosphere at a slow rate from 7.1 ± 0.2 ppt at the TP to 6.1 ± 0.5 ppt in the $LS^{top}$ layer (refer to Table 4). The tropospheric $CH_3Br$ (7.3 ± 0.2 ppt) is slightly larger than the 2016 global annually averaged 6.8 ppt as reported in Engel and Rigby et al. (2018), but agrees well with the NH measurements of $\sim$ 6.75–7.75 ppt from NOAA measurements which show that NH $CH_3Br$ is always larger than in the southern hemisphere (SH) (Fig. 1-7 of Engel and Rigby et al. (2018)). Contribution (2) to $Br^{tot}$, the four different halons (H–1301, H–1211, H–2402, and H–1202), has a mean VMR of 7.5 ± 0.2 ppt at the TP (in good agreement with the annually averaged tropospheric 7.8 ppt as from 2016 reported by WMO (2018)) and also decreases slowly to 6.5 ± 0.5 ppt by the $LS^{top}$ layer. The larger part of the $Br^{org}$ decrease in the LS is by contribution (3), the VSLS ($CH_2Br_2$, $CHBr_3$, $CH_2BrCl$, $CHBrCl_2$, and $CHBr_2Cl$), due to their short global lifetime of less than 6 months. At the TP, the $Br^{VSLS}$ is 3.1 ± 0.9 ppt (up to 7.2 ppt) while in the $LS^{top}$ layer the $Br^{VSLS}$ decreases to 1.0 ± 0.3 ppt. As such, the accumulated decrease of $Br^{org}$ (contributions (1) – (3) to $Br^{tot}$) throughout the LS between the TP and $\Delta\Theta$=88 K above the TP is $\sim$ 4.3 ppt.

Although further into the stratosphere the $Br^{org}$ is destroyed, bromine is converted to $Br_y^{inorg}$ and is compensated for in the $Br^{tot}$ budget. Contribution (4) to $Br^{tot}$, the $Br_y^{inorg}$, near the TP is around 1.5 ± 0.6 ppt (with a range of 0.2–3.3 ppt) and increases to 5.8 ± 1.8 ppt (2.9–10.1 ppt) within the $LS^{top}$ (refer to Table 4).”

Table 4. The average bromine mixing ratios of $CH_3Br$, halons, $Br^{VSLS}$, $Br^{org}$ ($CH_3Br$+halons+$Br^{VSLS}$), and $Br_y^{inorg}$ near the TP ($\Delta\Theta$ = ± 5 K) and in the upper layer of the lower stratospheric WISE measurements labeled $LS^{top}$ ($\Delta\Theta$ = 78-88 K).

| $\Delta\Theta$ from TP | $CH_3Br$ | Halons | $Br^{VSLS}$ | $Br^{org}$ | $Br_y^{inorg}$ |
|---|---|---|---|---|---|
| $LS^{top}$: $\Delta\Theta$=78-88 K | 6.1 ± 0.5 ppt | 6.5 ± 0.5 ppt | 1.0 ± 0.3 ppt | 13.5 ± 1.2 ppt | 5.8 ± 1.8 ppt |
| TP: $\Delta\Theta$= ± 5 K | 7.1 ± 0.2 ppt | 7.5 ± 0.2 ppt | 3.1 ± 0.9 ppt | 17.8 ± 1.2 ppt | 1.5 ± 0.6 ppt |

**Section 4.3.1, Pages 17-19: the fractions of air values for various domains of the atmosphere are presented and discussed. Would it be possible to give an estimate of the uncertainties corresponding to these values? For instance, in the investigated high bromine region, the fraction of air from the tropical troposphere is 51.2% while the corresponding value in the lower stratosphere below and above the HBrR is 42.6%. How significant is this difference?**

AC: The standard deviations of the CLaMS emission tracer fractions have been included as uncertainties in lines 553-564 for each origin region and two sentences have been added to explain their significance in lines 560 and 562-564:

“In the HBrR (marked by the black box), the fraction of air from the tropical troposphere is 51.2±8.3%, while only 27.1±8.1% is from the stratospheric background and smaller contributions are from the lower TTL (7.5±1.4%), mid-latitudinal LMS (7.0±0.6%), the mid-latitudinal troposphere (6.4±1.4%) and < 1% from the other domains (see also Fig. 11 (a)). In contrast, the air masses in the LS below and above

the HBrR (ranging from equivalent latitudes 55–80◦ N between the TP and Θ=350 K and Θ > 385 K) consist of 42.6±9.7% air from the tropical troposphere, 36.1±10.3% from the stratospheric background, 8.3±1.4% from the lower TTL, 7.0±0.6% from the mid-latitudinal LMS, 5.2±2.4% from the mid-latitudinal troposphere and < 1% from the other domains (see Fig. 11 (b)). The uncertainties stated (1σ standard deviation) indicate the variety of air masses mixed together in each domain.

Over the five months of CLaMS simulation prior to the HALO measurements, the influx of tropospheric air to the HBrR is almost 10% more than the surrounding LS below and above the HBrR. Although this difference is near the 1σ variability, the distributions of air mass origins from the HBrR compared to the LS below and above have distinct shifts especially for the tropical and mid-latitudinal troposphere and stratospheric background domains (not shown)."

The larger standard deviations indicate the variety of air masses being mixed together during this season. One thing to note from the attached figure below is that the distributions of measurements in the HBrR vs the LS below and above have a distinct difference and shift in relative contributions between them although the distributions overlap. This is most notable for the contributions from tropical troposphere, mid-lat troposphere, and stratospheric background. Also the distributions from the upper TTL and polar LMS are noticeably shifted but have significantly smaller contribution fractions overall.

[Figure]

**Page 24, line 770: Could simulations using the CLaMS model give indication about the seasonality of the bromine content in the high bromine region investigated in this study?**

AC: The current CLaMS model only includes the chemistry of inorganic bromine, therefore the high bromine region is not as prominent/noticeable. Ongoing current updates will include chemistry of organic bromine species for future CLaMS trace gas simulations. In principle CLaMS simulations could indicate the seasonality of bromine or other species if long term simulations are done in future studies.

Technical corrections:
**Page 6, line 176: 'asent' -> 'ascent'**
AC: This typo has accordingly been corrected for.

**Page 10, line 292: 'distince' -> 'distinct'**
AC: This typo has been corrected.

Manuscript number acp-2021-202 by Rotermund et al.
Reviewer 2, Qing Liang: June 16, 2021

We are very grateful for Qing Liang's positive comments and feedback and the recommendation for publication. We have included her suggestions for further improvements of our manuscript in the following way.

The reviewer's comments are written in **bold**, our responses are marked with AC (authors comments).
Comments:
**Would it be better to say "aged air" instead of "former air"?**
AC: In the abstract (line 11) the wording should imply that the air is still young and was recently in the troposphere rather than "aged" air. The sentence has been rephrased to avoid misinterpretations to read: "The most probable source region is air recently transported from the tropical upper troposphere and tropopause layer (UT/TTL) …".

**"one min time resolution" ï𝑓 one minute temporal resolution**
AC: The wording in line 285 has been changed as suggested to read: "one minute temporal resolution".

**Suggest changing to "nine distinct 3-D domains in the entire model atmosphere: the tropical troposphere, …**
AC: As suggested in line 295, the sentence to has been adjusted as follows: "nine distinct 3-D domains in the entire model atmosphere: the tropical troposphere, …".

**L300 & L599. Suggest change south-eastern Asia to Southeast Asia**
AC: We will continue to use the terminology of "south-eastern Asia" for two reasons. (1) The region we are referring to include the ocean and coastal areas rather than a specifically defined region of Southeast Asia landmasses. And (2) in figure 9, one of the sub-regions of the CLaMS model is labelled Southeast Asia (SEA), and when we are referring to south-eastern Asia it includes SEA but is not limited to that one region.

**Suggest change to "corresponding O3 loss"**
AC: As suggested the phrase "corresponding loss in $O_3$" in line 311-312 has been changed to: "corresponding $O_3$ loss.".

**Suggest replacing "off-line model" with TOMCAT model.**
AC: In line 312 we have changed the wording as suggested to: "Similarly to CLaMS, the TOMCAT model is driven…".

**Here, you assume everyone reads this paper knows about equivalent latitude, which is likely not the case with most people. I suggest you add a few sentences here explaining "what is equivalent latitude? What is the benefit of analysis in equivalent latitude coordinate?" for the benefits of general audience.**
AC: After the first mention of equivalent latitudes, we have included the following sentence for clarification in line 351: "… tropics (~6˚S) up to 86˚N. The equivalent latitude describes an enclosed area relative to the area of the globe of specific potential vorticity on a given potential temperature contour and is a useful quasi-Lagrangian coordinate for the interpretation of tracers in the stratosphere (Pan et al., 2012). The majority of the flight time …".

**L391 & L393. I suggest you add here the potential temperature range you would define as UTLS and LS. (You did mention the range of LS in L407, but it makes more sense to move it here). It might also be helpful to add horizontal lines on Figure 4 for clear identification of these regimes. Figure 4, please make the black solid line thicker so that it is easy to see.**

AC: In the corresponding lines 402-404, we have included the regions of the UTLS and LS in potential temperature difference from the tropopause as in Figure 4: "The weighted mean $Br^{tot}$ in the UTLS (all measurements, i.e., between $\Delta\Theta$ = -50K and 88 K) is 19.4 ± 1.3 ppt (the uncertainty being the $1\sigma$ variability of the measurements). Excluding the tropospheric measurements, the LS ($\Delta\Theta$ = 0-88 K) has a weighted average of …".

Figure 4 has also been updated as suggested: the black line has been made thicker, and gray shading has been added to identify the TP region ($\Delta\Theta$ = ±5 K) and the $LS^{top}$ layer ($\Delta\Theta$ = 78-88 K).

**L423-424. Is it fair to conclude based on your WISE analysis of in situ measurements that Br(tot) remain approximately constant vertically, implying as altitude increases, there are little change (loss) in total Br, but rather a gradual conversion from organic Br to inorganic Br? To me, this is actually a very important message for the Bromine community.**

AC: We have reworded the following sentence corresponding to the new lines 435-437 to change the emphasis as suggested: "The gradual conversion of ~ 4.3 ppt $Br^{org}$ to $Br_y^{inorg}$ by the $LS^{top}$ layer consequently results in minimal (or essentially no) changes of the $Br^{tot}$ budget, which is therefore approximately constant for the entirety of the air masses probed during the WISE campaign."

**I think "Impact on LMS O3" might be a better title. Consequences to me has a bit of negative tone.**

AC: The title of section 4.4 has been changed as suggested to "Impact on LMS $O_3$".

**You may consider redefine some of the acronyms here for those readers who come straight to the Conclusions section, e.g. NH, UTLS, LMS.**

AC: As recommended we have redefined all of the acronyms in the conclusion for completeness such that the conclusion can stand alone.

In line 744-745 "NH UTLS" has been written out as "northern hemisphere (NH) upper troposphere and lower stratosphere (UTLS)" for clarification.

In line 747 the term "LMS" is redefined as: "lowermost stratosphere (LMS)".

In line 752 the term "LS" is written out as: "lower stratosphere (LS)".

In line 754-755 "Ex-LS" is expanded to read: "extratropical lower stratosphere (Ex—LS)".

In line 759 "HBrR*" is redefined as: "a subsection of the high bromine region (HBrR*)".

In line 770-771 "tropical UT/TTL" is rewritten as: "tropical upper troposphere and tropopause layer (UT/TTL)".

In line 777 the phrase around "$\Theta_{high}/PV_{low}$" is reworded to describe the region: "into the tropical UT, i.e., the $\Theta_{high}/PV_{low}$ region,".

In line 780 the term "STE" is written out as: "stratosphere--troposphere exchange (STE)".

In line 785 the term "TP" is redefined as: "tropopause (TP)".